# Cis-regulatory evolution integrated the Bric-à-brac transcription factors into a novel fruit fly gene regulatory network

Maxwell J Roeske[1†], Eric M Camino[1†], Sumant Grover[1], Mark Rebeiz[2], Thomas Michael Williams[1,3]*

[1]Department of Biology, University of Dayton, Dayton, United States; [2]Department of Biological Sciences, University of Pittsburgh, Pittsburgh, United States; [3]Center for Tissue Regeneration and Engineering at Dayton, University of Dayton, Dayton, United States

**Abstract** Gene expression evolution through gene regulatory network (GRN) changes has gained appreciation as a driver of morphological evolution. However, understanding how GRNs evolve is hampered by finding relevant *cis*-regulatory element (CRE) mutations, and interpreting the protein-DNA interactions they alter. We investigated evolutionary changes in the duplicated Bric-à-brac (Bab) transcription factors and a key Bab target gene in a GRN underlying the novel dimorphic pigmentation of *D. melanogaster* and its relatives. It has remained uncertain how Bab was integrated within the pigmentation GRN. Here, we show that the ancestral transcription factor activity of Bab gained a role in sculpting sex-specific pigmentation through the evolution of binding sites in a CRE of the pigment-promoting *yellow* gene. This work demonstrates how a new trait can evolve by incorporating existing transcription factors into a GRN through CRE evolution, an evolutionary path likely to predominate newly evolved functions of transcription factors.
DOI: https://doi.org/10.7554/eLife.32273.001

*For correspondence:
twilliams2@udayton.edu

†These authors contributed equally to this work

Competing interests: The authors declare that no competing interests exist.

## Introduction

Transcription factors play central roles in the development and evolution of animal traits by binding to *cis*-regulatory elements to spatially and temporally regulate patterns of gene expression (*Davidson, 2006*; *Levine, 2010*). Collectively, the complex web of connections between transcription factors and CREs form vast gene regulatory networks (GRN) that govern a tissue's development. As transcription factor genes are generally much older than the traits they impact, a central question of evolutionary developmental biology is the relative role that gene duplication, protein coding and CRE sequence evolution play in the evolution of GRNs for novel traits (*Carroll, 2008*; *Stern and Orgogozo, 2008*). Answers to this question require studies of recently evolved traits for which the derived and ancestral states of genes and gene components can be inferred through manipulative studies of GRN function (*Rebeiz and Williams, 2011*).

One such experimentally tractable trait is the rapidly evolving pigmentation patterns that adorn the abdominal cuticle of *Drosophila melanogaster* and its close relatives (*Rebeiz and Williams, 2017*). The melanic pigmentation of the dorsal cuticle tergites covering the A5 and A6 abdominal segments of males has been inferred to be a novelty that evolved in the *D. melanogaster* lineage after it diverged from an ancestral monomorphically pigmented lineage within the *Sophophora* subgenus (*Jeong et al., 2006*). Within the GRN controlling this trait, the Hox gene *Abd-B* plays an important role in activating the expression of terminal enzyme genes that are required for pigment formation (*Figure 1*). One enzyme, encoded by the *yellow* gene is activated through a CRE known as the 'body element' (*Camino et al., 2015*; *Wittkopp et al., 2002*) that possesses at least two

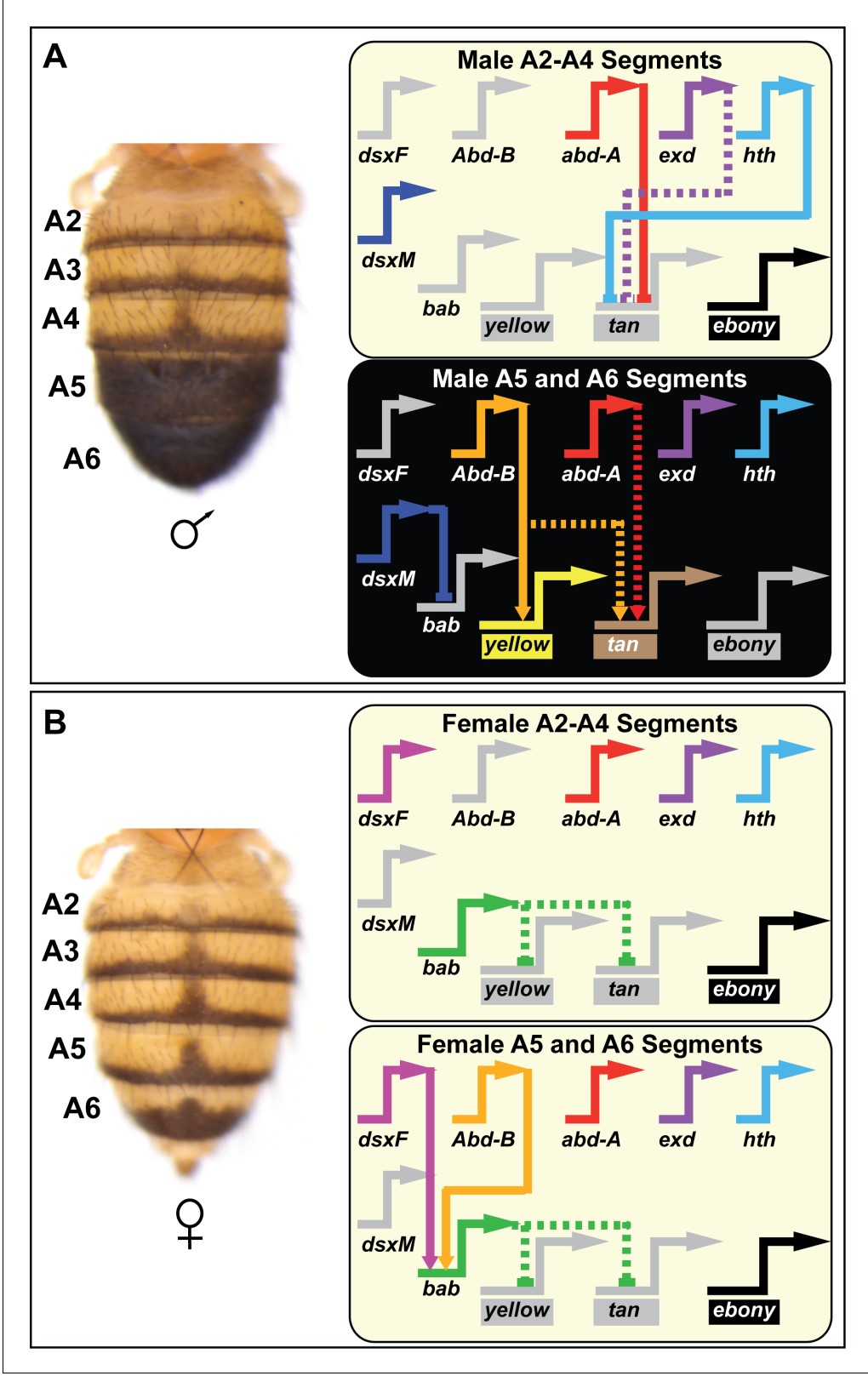

**Figure 1.** Contemporary model for the *D. melanogaster* tergite pigmentation Gene Regulatory Network. Melanic tergite pigmentation requires the specific expression of the pigmentation genes *yellow* and *tan*, while blocking the expression of the yellow-pigment promoting *ebony* gene. (**A**) Tergite pigmentation pattern for a *D. melanogaster* male and the regulatory interactions experienced in the non-melanic male A2-A4 segments (top)

*Figure 1 continued on next page*

*Figure 1 continued*

and the melanic A5 and A6 segments (bottom). *Abd-B* is not expressed in the anterior A2-A4 segments and resultantly *yellow* and *tan* lack the direct and indirect activating input. In these anterior segments, Abd-A acts as a direct repressor of *tan* in combination with (direct or indirect) repressive effects of *exd* and *hth*. Abd-B is expressed in the posterior A5 and A6 segments, where it functions as a direct activator of *yellow* and an indirect activator of *tan*. In these segments, Abd-A acts as an indirect activator of *tan*. (B) Tergite pigmentation pattern of a *D. melanogaster* female and the key regulatory inputs experienced in the A2-A4 segments (top) and the A5 and A6 segments (bottom). In the female abdomen, Bab acts as a dominant repressor of *yellow* and *tan* expression, overriding the presence of Abd-B and Abd-A. In the GRN schematics, inactive genes are indicated in gray coloring, solid lines connecting genes indicate established direct interactions between a transcription factor and a target gene's CRE, and dashed connections indicate indirect regulatory interactions or interactions not yet shown to be direct. Lines terminating with an arrowhead indicate regulation in which the transcription factor functions as an activator, and lines terminating in a nail-head shape indicate repression.

DOI: https://doi.org/10.7554/eLife.32273.002

binding sites for the Hox factor Abd-B (*Jeong et al., 2006*). In *D. melanogaster* females, *yellow* expression is repressed in the A5 and A6 segments through its regulation by the tandem duplicate *bab1* and *bab2* genes (*Figure 2A*) (*Jeong et al., 2006*), collectively referred to as Bab. Both paralogs encode proteins that possess a conserved BTB domain that functions in homodimerization and heterodimerization, and a conserved domain with pipsqueak (psq) and AT-hook domains that together confer a similar in vitro DNA-binding capability (*Lours et al., 2003*). In contrast to the 78% amino acid identity observed for both BTB and conserved domain of the paralogs, amino acid identity drops to ~26% throughout the remainder of these proteins (*Couderc et al., 2002*) (*Figure 2— source data 1*). Elsewhere, it was reported that a high amino acid identity exists between the conserved domains of the *D. melanogaster* paralogs and the Bab orthologs for the mosquito *A. gambiae* and honeybee *A. mellifera*, and most of the amino acid differences being conservative (*Lours et al., 2003*). Although the Bab proteins are suspected to function as transcription factors, no direct targets of regulation are known (*Jeong et al., 2006*).

In the lineage of *D. melanogaster*, Bab expression is suspected to have evolved from a monomorphic pan-abdominal pattern to a sexually dimorphic expression pattern in which both paralogs are female-limited, and absent from males during the latter half of pupal development when enzymes of the pigmentation GRN are deployed (*Kopp et al., 2000*; *Salomone et al., 2013*). This dimorphic pattern of Bab expression required changes to two CREs controlling Bab's abdominal epidermis expression (*Williams et al., 2008*). Although this dimorphic pattern of regulation allows *yellow* to be expressed in the epidermis underlying the pigmented male A5 and A6 segment tergites, several questions remain unanswered pertaining to how and when Bab was incorporated into the sexually dimorphic pigmentation GRN. Specifically, is *yellow* a direct target of Bab repression, and was it a target of regulation prior to the evolution of the dimorphic trait (*Gompel and Carroll, 2003*)? Moreover, to what extent did gene duplication, protein coding sequence, and CRE evolution contribute to this derived trait?

In this study, we sought to characterize the derived functions of Bab1 and Bab2 in the *D. melanogaster* pigmentation GRN and determine the extent to which this derived pigmentation function additionally required evolutionary changes to Bab protein coding sequences and the 5' *cis*-regulatory region of *yellow* that contains its abdominal CRE. We found that Bab1 and Bab2 bind directly to the *yellow* body CRE to a region required for male-limited enhancer activity. Moreover, the capability of Bab paralogs to function in this repressive manner appears to result not from protein-coding changes, but instead through the evolution of these binding sites which arose contemporaneously with the derived pattern of dimorphic Bab expression. Thus, the origin of the male-specific pigmentation of *D. melanogaster* is an example where evolutionarily conserved transcription factors gained a new function through their integration into a GRN by CRE evolution.

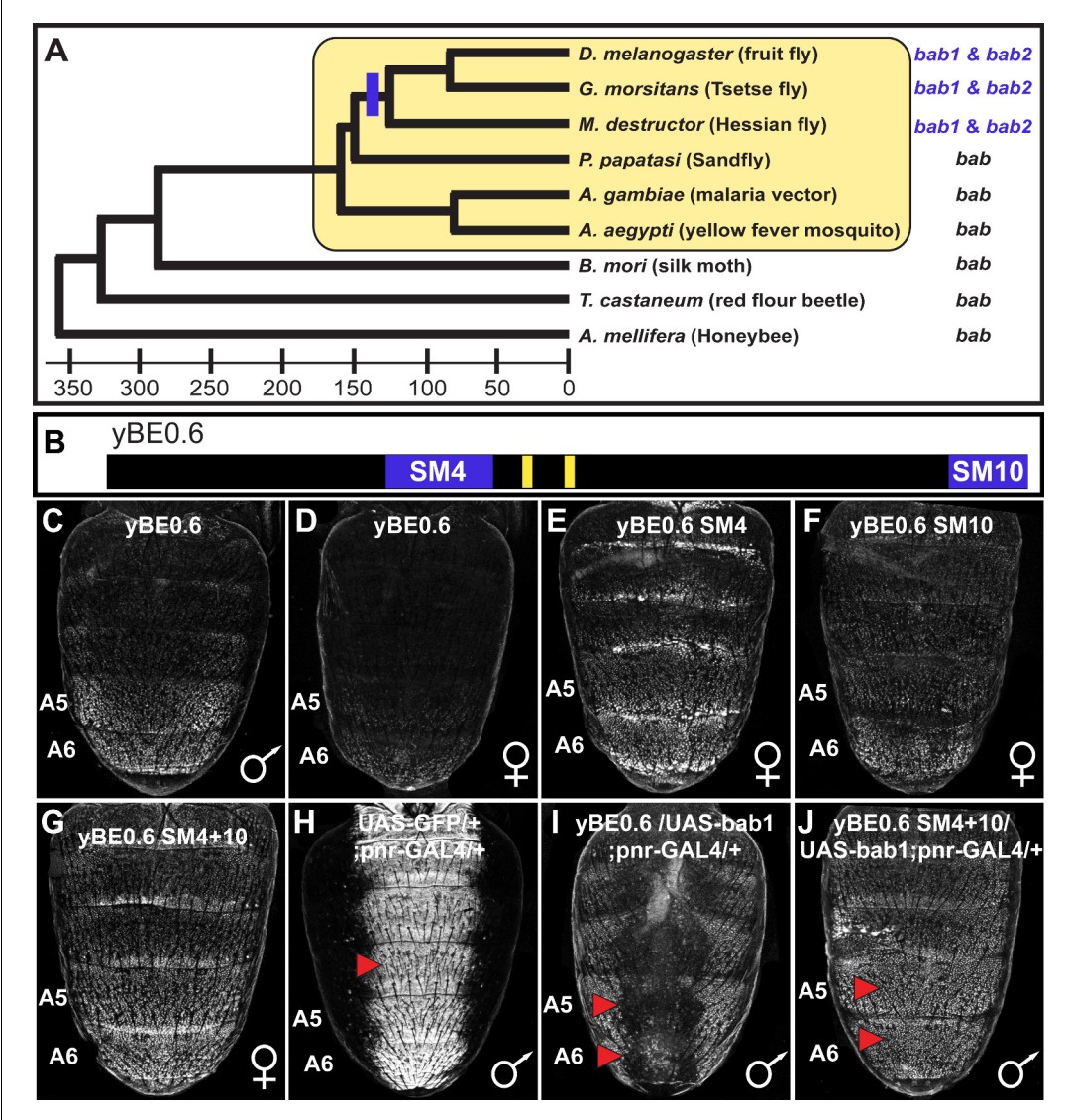

**Figure 2.** The tandemly duplicated *bab* genes perform a derived role in repressing a CRE controlling male-specific expression of the gene *yellow*. (A) An ancestral *bab* gene was duplicated into the paralogous *bab1* and *bab2* genes in a Dipteran lineage that includes *Drosophila* fruit flies. The time scale indicates approximate divergence times in millions of years ago. (B) Male-specific expression of *yellow* in the abdominal epidermis is under the control of the yBE0.6 CRE that possesses two binding sites for Abd-B that are shown as yellow rectangles. Blue bars delimit the SM4 and SM10 regions required to suppress CRE activity in females. (C and D) The yBE0.6 EGFP reporter transgene is elevated in the male A5 and A6 abdomen segments (C) but is only barely detected females (D). (E–G) Ectopic reporter expression occurs in the female abdomen when either the SM4, SM10, or both regions are mutated. (H) The *pnr*-GAL4 driver activates dorsal midline expression of the UAS-EGFP gene, demarcating its domain of misexpression. (I) Dorsal midline expression of the yBE0.6 CRE is lost when *bab1* is ectopically expressed by *pnr*-GAL4. (J) When the SM4 and SM10 regions are mutated, the yBE0.6 CRE can activate reporter expression in midline regions in spite of ectopically expressed *bab1*.
DOI: https://doi.org/10.7554/eLife.32273.003

The following source data and figure supplement are available for figure 2:

**Source data 1.** Amino acid alignment for Bab homologs.
DOI: https://doi.org/10.7554/eLife.32273.005

**Source data 2.** Sequence alignment of the yBE0.6 with scanning mutant versions.
DOI: https://doi.org/10.7554/eLife.32273.006

**Figure supplement 1.** Scanning mutagenesis across the entire yBE0.6 CRE identifies sequences that normally function to repress CRE activity in the female abdomen.
DOI: https://doi.org/10.7554/eLife.32273.004

## Results

### Bab1 suppresses *yellow* expression through *cis*-regulatory element encodings

We sought to characterize how Bab1 exerts its influence on a minimal 0.6 kb body element CRE (yBE0.6) (*Camino et al., 2015*) (*Figure 2B*) that drives male-limited GFP reporter transgene expression in the dorsal epidermis of the A5 and A6 abdominal segments (*Figure 2C and D*). This reporter transgene activity matches the spatial, sex-limited, and temporal pattern of abdominal expression of *yellow* (*Camino et al., 2015*). We designed a set of 10 mutant yBE0.6 CREs (*Figure 2—source data 2*) to localize regions responsible for this element's sex-limited activity. In each mutant yBE0.6, we introduced a block of ~70–85 base pairs in which every other base pair possessed non-complementary transversion mutations and compared its activity to the wild type CRE in transgenic *D. melanogaster* (*Camino et al., 2015*). While 8 of 10 'scanning mutant' CREs showed wild type reporter repression in the female abdomen (*Figure 2—figure supplement 1*), we observed increased expression in the A5 and A6 segments of the SM4 and SM10 mutants (*Figure 2E and F*). Moreover, the increased expression was more pronounced when the SM4 and SM10 mutations were combined in the same reporter (*Figure 2G*). These results identified two CRE sub-regions that are required to suppress *yellow* expression in the posterior female abdomen, likely through the recruitment of a transcriptional repressor protein.

We speculated that the sequences altered by the SM4 and SM10 mutations normally function to respond either directly or indirectly to the repressive Bab proteins. When Bab1 was ectopically expressed in the dorsal midline of the male abdomen by the GAL4/UAS system (*Brand and Perrimon, 1993*; *Calleja et al., 2000*), yBE0.6 reporter expression was largely suppressed (*Figure 2H and I*). However, a yBE0.6 CRE containing the SM4 and SM10 mutations, was unresponsive to ectopic Bab1 expression (*Figure 2J*). These data reveal that the SM4 and SM10 CRE regions encode inputs that respond to regulation by Bab proteins.

### Bab1 directly interacts with multiple *yellow* CRE binding sites

Bab proteins may suppress the yBE0.6 CRE activity through two major routes: indirect or direct regulations. Bab may indirectly regulate the yBE0.6 CRE by controlling the expression of a transcription factor that interacts with binding sites in the SM4 and SM10 regions. On the other hand, Bab may directly interact with binding sites in the yBE0.6 CRE, acting as a transcriptional repressor. To distinguish between these mechanisms, we performed gel shift assays to see whether Bab specifically interacts with sequences within the SM4 and SM10 regions in vitro. Previously, it was shown that the DNA-binding domain (DBD) of Bab1 and Bab2 bound to similar DNA sequences (*Lours et al., 2003*); therefore, we chose to perform our experiments with the Bab1 DBD.

The SM4 region is 70 base pairs (bp) in length (*Figure 3A*), which we divided into three overlapping sub element regions that were tested for binding by serial two-fold dilutions of a GST fusion protein possessing the Bab1 DNA-binding domain (Bab1 DBD, *Figure 3B*). Of the three regions, only the third probe was substantially bound by the Bab1 DBD (*Figure 3C–E*). This binding was DNA-sequence specific, as a scanning mutant probe version failed to similarly shift (*Figure 3F*). Thus, it seems likely that this 25 base pair segment possesses a site or sites capable of Bab1 DNA-binding.

A previous study showed that the Bab1 DBD preferably bound A/T-rich sequences, specifically those with TAA or TA repeats (*Lours et al., 2003*). Within region three are several TA motifs, several of which are part of TAA motifs (*Figure 3A*). We created a TA > GA mutant probe that removed each TA motif. This probe was not noticeably bound by the Bab1 DBD indicating that some or all these motifs are necessary features of a Bab1 binding site or sites (*Figure 3G*). To further localize the sequences necessary for the Bab1 binding, we created four mutant probes within the region (sub1-sub4). We found that three of these mutant probes were still bound and shifted by the Bab1 DBD (*Figure 3H–K*). However, the sub2 probe that spanned nine base-pairs and disrupted two of the TA motifs was not noticeably shifted (*Figure 3I*). Collectively, this set of gel shift assays supports a direct Bab1 binding mechanism to suppress this CRE sequence (*Figure 3A*, red sequence).

We performed a similar set of gel shift assays to localize Bab1 binding within the 85 bp SM10 region (*Figure 4*). We tested three overlapping sub element regions for an interaction with the Bab1

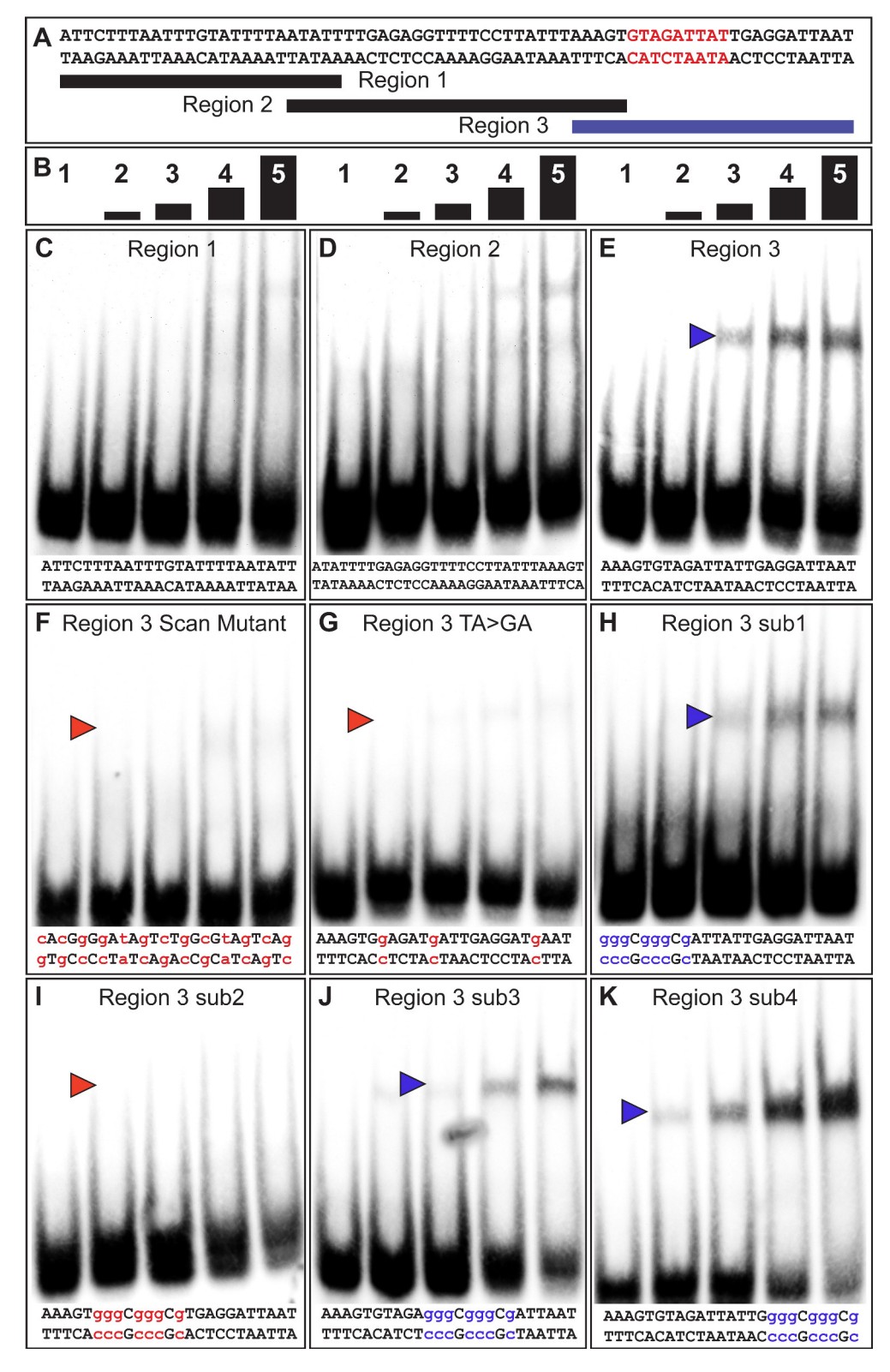

**Figure 3.** The yBE0.6 possesses a binding site for Bab in the SM4 region. (**A**) The wild type DNA sequence of the SM4 region is shown, which was subdivided into three smaller regions annotated below that were used as double stranded probes in gel shift assays with the GST-Bab1 DNA-binding Domain (Bab1-DBD). Red text delimits the inferred Bab-binding site. (**B**) Each probe was tested in gel shift assay reactions for binding with five different amounts of Bab1-DBD. These were from left to right: 0, 500, 1000, 2000, and 4000 ng. (**C–E**) Gel shift assays using wild type probe sequences. (**F–K**) Gel

*Figure 3 continued*

shift assays using mutant probe sequences. Lower case blue letters indicate probe mutations that did not noticeably alter protein binding. Probe base pairs in lower case red letters are changes that altered protein binding. Blue and red arrowheads indicate the location of shifted probe, with red arrowheads indicating cases where the quantity of shifted probe was noticeably reduced.

DOI: https://doi.org/10.7554/eLife.32273.007

DBD (*Figure 4C–E*), and observed mobility shifts for the first and third regions (*Figure 4C and E*). A scanning mutant version of the third region was bound, albeit to a lesser extent, by Bab1 (*Figure 4G*), indicating that this in vitro binding was not sequence-specific or that the mutant probe created a weak Bab-binding site. Therefore, we focused our attention on the first region for which a scan mutant probe was not noticeably bound by the Bab1 DBD (*Figure 4F*). We next analyzed four small mutant versions (sub1-sub4) of the first region (*Figure 4H–K*). While each mutant probe appeared to be bound to a lesser extent than the wild type probe, the sub3 mutant probe showed little-to-no binding (*Figure 4J*). These results suggest the presence of at least one additional Bab-binding site in the SM10 region. Collectively, our results support a direct model of regulation in which Bab1 functions as a DNA-binding transcription factor that interacts with at least two sites within the yBE0.6 CRE and thereby represses the expression of this CRE in the female abdomen where Bab1 is highly expressed.

## The biochemical activity of Bab predates its duplication event

While the *bab* genes have been shown to be sufficient to suppress *D. melanogaster* tergite pigmentation when ectopically expressed (*Couderc et al., 2002*; *Kopp et al., 2000*), the individual necessities of *bab1* and *bab2* paralogous genes have not been fully resolved. We created two short hairpin/miRNA (shmiR) transgenic lines that can conditionally and specifically target sequences unique to *bab1* (*Table 1*) and separately two lines targeting *bab2* (*Table 2*) for RNA-interference (RNAi). These shmiR transgenes are under the *cis*-regulatory control of Upstream Activating Sequences (UAS), and thus expression can be induced by the GAL4 transcription factor (*Haley et al., 2008*). Using a GAL4 insertion into the *pannier* (*pnr*) gene, we drove hairpins specific to a negative control gene (targeting the *mCherry* reporter gene) and to either or both *bab1* and *bab2* along the dorsal midline of the body. Relative to the control (*Figure 5A*), the *bab1* shmiR transgene containing the siRNA id #3 sequence (*Table 1*) led to a conspicuous increase in the dorsal medial pigmentation of the female A5 and A6 tergites (*Figure 5B*), whereas the transgene including the siRNA id #4 sequence (*Table 1*) resulted in a phenotype not noticeably different from that of the negative control (*Figure 5C*). For *bab2*, the individual ectopic expression of the shmiR transgenes containing either the siRNA id #12 or #16 sequences (*Table 2*) resulted in ectopic dorsal medial tergite pigmentation in females (*Figure 5D and E*). Hence, these results demonstrate that suppression of tergite pigmentation in *D. melanogaster* females requires individual contributions from both *bab1* and *bab2*.

The RNAi knockdown of *bab1* and *bab2* each increased pigmentation, suggesting that we would obtain a more expressive phenotype if both paralogs' expression were suppressed. To test this prediction, we created 'chained' shmiRs (*Haley et al., 2010*) to co-express the effective *bab1* shmiR (Id #3) separately with each *bab2* shmiR (Id #12 and #16). We found that these chained shmiR transgenes, when ectopically expressed in the dorsal midline resulted in more expansive ectopic pigmentation phenotypes (*Figure 5F and G*). Notably, for one chained combination, ectopic pigmentation included the A4 tergites of males and females (*Figure 5F*, red arrowheads). These results show that *bab1* and *bab2* paralog expression are necessary to fully suppress tergite pigmentation, including the A4 segment of males and females.

Not surprisingly, the RNAi phenotypes were most extreme in the female abdomen as the pupal male abdomen lacks significant *bab1* and *bab2* expression (*Salomone et al., 2013*). Previous studies have shown that both *bab1* and *bab2* are sufficient to suppress male tergite pigmentation when ectopically expressed (*Couderc et al., 2002*; *Kopp et al., 2000*). However, direct comparisons of the individual paralogs were hampered by the positional effects associated with random insertion of *bab* paralog open-reading frame (ORF) transgenes into different genomic sites (*Couderc et al., 2002*; *Kopp et al., 2000*). Here, we created transgenes with the *D. melanogaster bab1* and *bab2* ORFs under UAS regulation that were integrated site-specifically into the attP40 site on the second

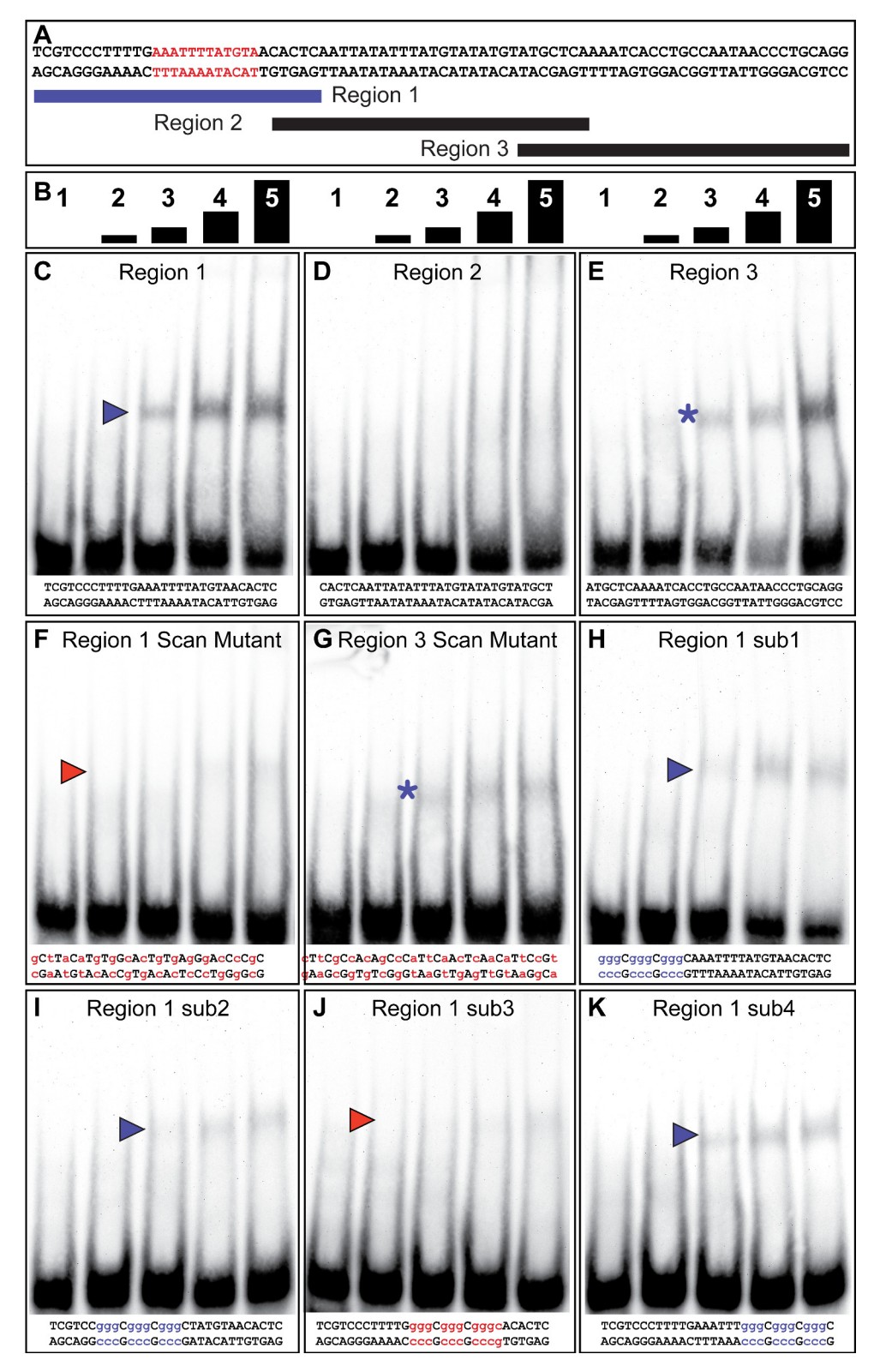

**Figure 4.** The yBE0.6 possesses a binding site for Bab in the SM10 region. (**A**) The wild type DNA sequence of the SM10 region is shown, which was subdivided into three smaller regions annotated below that were used as double stranded probes in gel shift assays with the GST-Bab1 DNA-binding Domain (Bab1-DBD). Red text delimits the inferred Bab binding site. (**B**) Each probe was tested in gel shift assay reactions for binding with five different amounts of Bab1-DBD. These were from left to right: 0, 500, 1000, 2000, and 4000 ng. (**C–E**). Gel shift assays using wild type probe sequences. (**F–K**)
*Figure 4 continued on next page*

*Figure 4 continued*

Gel shift assays using mutant probe sequences. Lower case purple letters indicate probe mutations that did not noticeably alter protein binding. Probe base pairs in lower case red letters are changes that altered protein binding. Purple and red arrowheads indicate the location of shifted probe, with red arrowheads indicating cases where the quantity of shifted probe was noticeably reduced. Asterisks indicate a situation where binding was non-specific as both the wild type and mutant probes were bound by the Bab1-DBD.

DOI: https://doi.org/10.7554/eLife.32273.008

chromosome. In the absence of a GAL4 driver, leaky expression of these transgenes resulted in reduced pigmentation of the male A5 and A6 tergites (*Figure 6A and B*). When we drove ectopic expression of these ORFs in the dorsal midline by the *pnr-GAL4* chromosome, we failed to recover viable adult males which possessed the UAS-*bab1* ORF transgene, indicating that the ectopic-expression phenotype was lethal. While we obtained fewer than expected offspring with ectopic expression of the *bab2* ORF than expected for independent assortment (*Figure 6—figure supplement 1*), some adults were identified. For these specimens, tergite pigmentation was eliminated in the dorsal midline of males, and a non-specific split tergite phenotype was seen for both males and females (*Figure 6—figure supplement 2*). These outcomes show that *bab1* and *bab2* have a strong pigment-suppressing capability, although in the genetic background tested, *bab1* ectopic expression was lethal.

The lethality encountered in the ectopic-expression assays likely stems from expressing the *bab* ORFs in the spatial and temporal pattern of the *pnr* gene (*Calleja et al., 2000*) which drives strong expression from embryonic stages through pupal development. In order to better visualize the specific effects of *bab* ORF expression on tergite pigmentation, we utilized the *y-GAL4* transgene (*Hart, 2013*) to drive expression in the abdominal epidermis pattern of the *yellow* gene which begins around 70 hours after puparium formation (hAPF) (*Figure 6—figure supplement 3*) based on a 100-hour period of pupal development (*Rogers and Williams, 2011*). Expression of the *bab1* and *bab2* ORFs driven by the *y-GAL4* chromosome eliminated both lethality and tergite developmental defects, resulting in male adults which entirely lacked melanic pigmentation on the A5 and A6 tergites (*Figure 6G and H*). These data show that Bab1 and Bab2 are both potent suppressors of pigmentation in two very different regimes of ectopic expression.

## Bab1 ectopic-expression phenotypes require DNA-binding capability

The Bab1 and Bab2 proteins possess a conserved domain that includes both pipsqueak (psq) and AT-Hook motifs that function as an in vitro DNA-binding domain or DBD (*Lours et al., 2003*), supporting the notion that these paralogs function as transcription factors in vivo. However, bona fide direct targets of either Bab1 or Bab2 have yet to be discovered. Previously, it was shown that the Bab1 DBD failed to bind DNA in vitro when possessing non-synonymous mutations in the pipsqueak

**Table 1.** Design of small interfering RNA output for the *bab1* ORF.

| siRNA id | Position | SS sequence (Passenger) | AS sequence (Guide) | Corrected score |
|---|---|---|---|---|
| 1 | 560 | GGAUAGCUGAGAUGUUGAAAG | UUCAACAUCUCAGCUAUCCUG | 99.7 |
| 2 | 1442 | CCGAUGACUUGGAGAUCAAGC | UUGAUCUCCAAGUCAUCGGCG | 85.7 |
| 3 | 278 | GGAACAACUAUCAGACGAACC | UUCGUCUGAUAGUUGUUCCAG | 97.6 |
| 4 | 162 | GAGUCAAGGUCAUGCUGUAGC | UACAGCAUGACCUUGACUCUC | 95.2 |
| 5 | 1483 | CGAGAGGAAGAAAGGGUAAGU | UUACCCUUUCUUCCUCUCGGA | 81.8 |
| 6 | 150 | CGAGGACAAGGAGAGUCAAGG | UUGACUCUCCUUGUCCUCGUC | 94.6 |
| 7 | 1473 | CGAGAUGAUCCGAGAGGAAGA | UUCCUCUCGGAUCAUCUCGGC | 78.9 |
| 8 | 219 | GGGCAGGAGUUCUUCGGUAGC | UACCGAAGAACUCCUGCCCUG | 89.5 |
| 9 | 359 | GCGAUGGUCGGUCCAUGAAGG | UUCAUGGACCGACCAUCGCAU | 88.2 |
| 10 | 664 | CCCAAGGAGAGCACUUCAACU | UUGAAGUGCUCUCCUUGGGCG | 84.7 |
| 11 | 368 | GGUCCAUGAAGGCCCACAAGA | UUGUGGGCCUUCAUGGACCGA | 87.5 |

DOI: https://doi.org/10.7554/eLife.32273.011

**Table 2.** Design of small interfering RNA output for the *bab2* ORF.

| siRNA_id | Position | SS sequence (Passenger) | AS sequence (Guide) | Corrected score |
|---|---|---|---|---|
| 6 | 16 | GAUUGUGGACUUUGAAAUAAA | UAUUUCAAAGUCCACAAUCUG | 98.1 |
| 12 | 279 | CGGAGCUGGUGAAGUCCAAGG | UUGGACUUCACCAGCUCCGUU | 94.5 |
| 20 | 51 | GCGAAAUCGAUCAGUUCGAGG | UCGAACUGAUCGAUUUCGCCG | 94.4 |
| 18 | 155 | AGAAAGUACUCACCCGAAAGG | UUUCGGGUGAGUACUUUCUGU | 93.6 |
| 19 | 202 | AAGUGAGGUGGUUGAUCAAAU | UUGAUCAACCACCUCACUUGG | 92.5 |
| 23 | 241 | CGUUGGAGAAGUCAAGUCACC | UGACUUGACUUCUCCAACGCU | 92.3 |
| 42 | 1 | GGACAUGACCAAACAGAUUGU | AAUCUGUUUGGUCAUGUCCAU | 91.7 |
| 43 | 14 | CAGAUUGUGGACUUUGAAAUA | UUUCAAAGUCCACAAUCUGUU | 91.6 |
| 45 | 63 | AGUUCGAGGCGAGUGACUACA | UAGUCACUCGCCUCGAACUGA | 91.4 |
| 40 | 154 | CAGAAAGUACUCACCCGAAAG | UUCGGGUGAGUACUUUCUGUU | 90.7 |
| 49 | 13 | ACAGAUUGUGGACUUUGAAAU | UUCAAAGUCCACAAUCUGUUU | 90.7 |
| 28 | 306 | CGAUGAACGACCAAGCUUUGA | AAAGCUUGGUCGUUCAUCGGA | 90.6 |
| 46 | 140 | CUAGAGGACCAGAACAGAAAG | UUCUGUUCUGGUCCUCUAGAU | 90.4 |
| 13 | 625 | GACCAAUGUCUUUGACGAACU | UUCGUCAAAGACAUUGGUCAG | 90.3 |
| 55 | 12 | AACAGAUUGUGGACUUUGAAA | UCAAAGUCCACAAUCUGUUUG | 90.3 |
| 38 | 297 | AGGCGAGUCCGAUGAACGACC | UCGUUCAUCGGACUCGCCUUG | 89.8 |
| 27 | 443 | CAGCCUCAACCAAAUCUUAAG | UAAGAUUUGGUUGAGGCUGUG | 89.6 |
| 10 | 822 | UGGUGGAGUUCAUGUACAAGG | UUGUACAUGAACUCCACCAGG | 88.9 |
| 4 | 1099 | GGACUUGAAUCAGCGACAAAG | UUGUCGCUGAUUCAAGUCCAA | 88.8 |

DOI: https://doi.org/10.7554/eLife.32273.012

(psq) motif converting Alanine and Isoleucine amino acids to Glycine and Proline, respectively (AI576GP), or when non-synonymous mutations altered a stretch of Arginine, Glycine, and Arginine amino acids in the AT-Hook motif to Aspartic acid, Glycine, and Aspartic acid (RGR627DGD), respectively (*Lours et al., 2003*). To test whether this compromised ability to bind DNA in vitro has in vivo significance, we created a *bab1* ORF transgene that possesses both the psq and AT-Hook mutations (called *bab1* DNA-binding mutant or *bab1* DBM, *Figure 6—source data 1*) and incorporated this transgene into the same genomic site as our other UAS transgenes. We found that leaky expression of the *bab1* DBM was insufficient to suppress tergite pigmentation (*Figure 6C*). Moreover, ectopic expression of the Bab1 DBM by the *pnr-GAL4* driver resulted in detectable accumulation of nuclear protein (*Figure 6—figure supplement 4*) which did not induce lethality or a pigmentation phenotype (*Figure 6—figure supplements 1* and *2*). Similarly, expression of the Bab1 DBM in the *y-GAL4* pattern resulted in males with the wild type melanic tergites (*Figure 6I*). Collectively, these results lend further support to a model in which DNA binding is required for the *D. melanogaster* Bab paralogs to repress tergite pigmentation.

## Functional equivalence of bab homologs for the suppression of tergite pigmentation

The currently favored model for the origin of the *D. melanogaster* sexually dimorphic tergite pigmentation posits that it evolved from an ancestor that expressed Bab in a sexually monomorphic manner, and for which melanic pigmentation in males and females was limited (*Jeong et al., 2006*; *Kopp et al., 2000*; *Salomone et al., 2013*). Moreover, CRE evolution has prominently factored into the origin of this dimorphic pigmentation trait, as changes in CREs of *bab* have been previously identified (*Williams et al., 2008*). The possibility that Bab protein coding sequence evolution has additionally contributed has largely remained untested. To investigate whether the Bab1 and Bab2 proteins have functionally evolved, we created UAS-regulated transgenes possessing the *D. willistoni bab1* and *D. mojavensis bab2* ORFs (*Figure 6—source data 1*). These orthologous protein coding sequences come from fruit fly species presumed to possess the ancestral sexually monomorphic

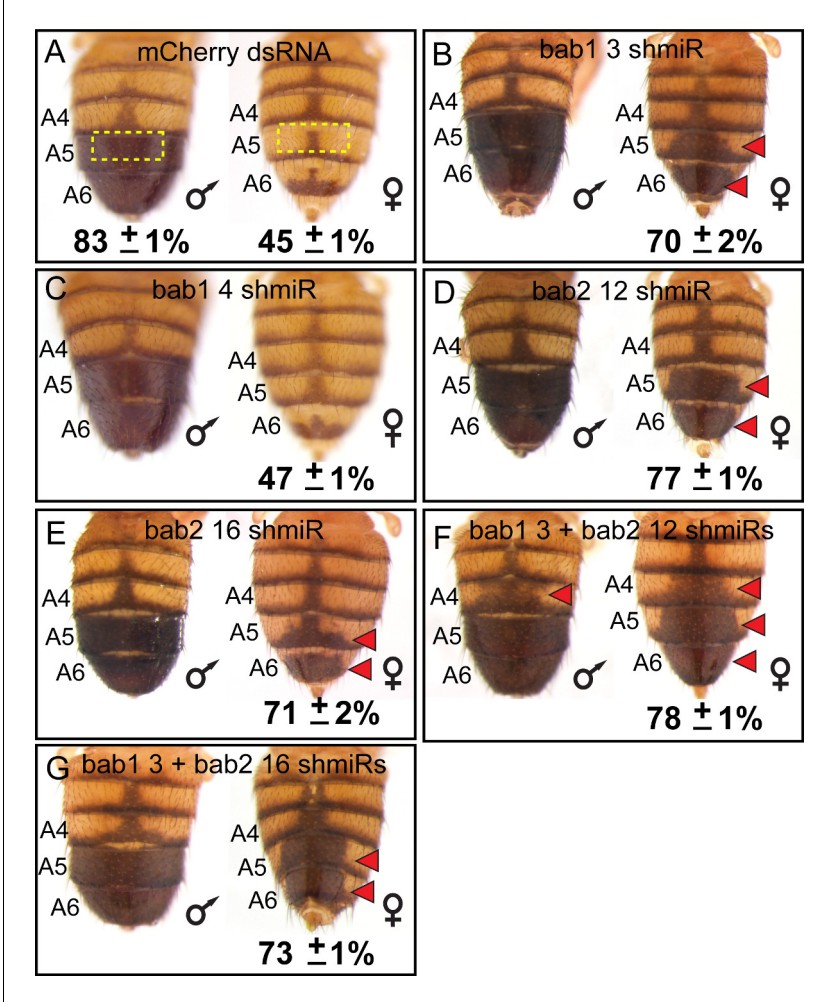

**Figure 5.** RNA-interference reveals a necessity for both *bab1* and *bab2* in suppressing female tergite pigmentation. (A–G) Double-stranded (ds) RNA transgenes with UAS binding sites were expressed in the dorsal midline abdomen region driven by GAL4 that was expressed in the midline pattern of the *pnr* gene. (A) Expression of a negative control dsRNA that targets a gene (*mCherry*) that does not naturally exist in the *D. melanogaster* genome resulted in no apparent pigmentation phenotype from RNA-interference (RNAi). (B and C) Two different dsRNAs specific to *bab1* and to (D and E) *bab2* were tested for pigmentation phenotypes from RNAi. (F and G) Simultaneous RNAi for *bab1* and *bab2* was accomplished by expressing 'chained' transgenes. Red arrowheads indicate tergite regions where RNAi caused the development of ectopic pigmentation. The anterior midline tergite regions (illustrated in panel A by dashed yellow rectangles) were quantified for their darkness percentage for replicate specimens (n = 4). These percentages and their standard error of the mean (±SEM) are provided below a representative image.

DOI: https://doi.org/10.7554/eLife.32273.009

The following source data is available for figure 5:

**Source data 1.** Analysis of RNA-interference pigmentation phenotypes.
DOI: https://doi.org/10.7554/eLife.32273.010

patterns of pigmentation and Bab expression. At the time this study was initiated, the *D. willistoni bab2* ORF included far fewer codons than the *D. melanogaster* ortholog, indicating that the full ORF remained uncharacterized. Thus, we opted to use the ORF for the *D. mojavensis* ortholog. As seen for the *D. melanogaster* ORF transgenes, leaky expression from the attP2 genomic site of transgene insertion resulted in a similar reduction in male A5 and A6 tergite pigmentation (*Figure 6D D. willistoni bab1*; and 6E, *D. mojavensis bab2*). Pigmentation was dramatically suppressed when these orthologous proteins were ectopically expressed in the dorsal medial midline pattern of *pnr*-

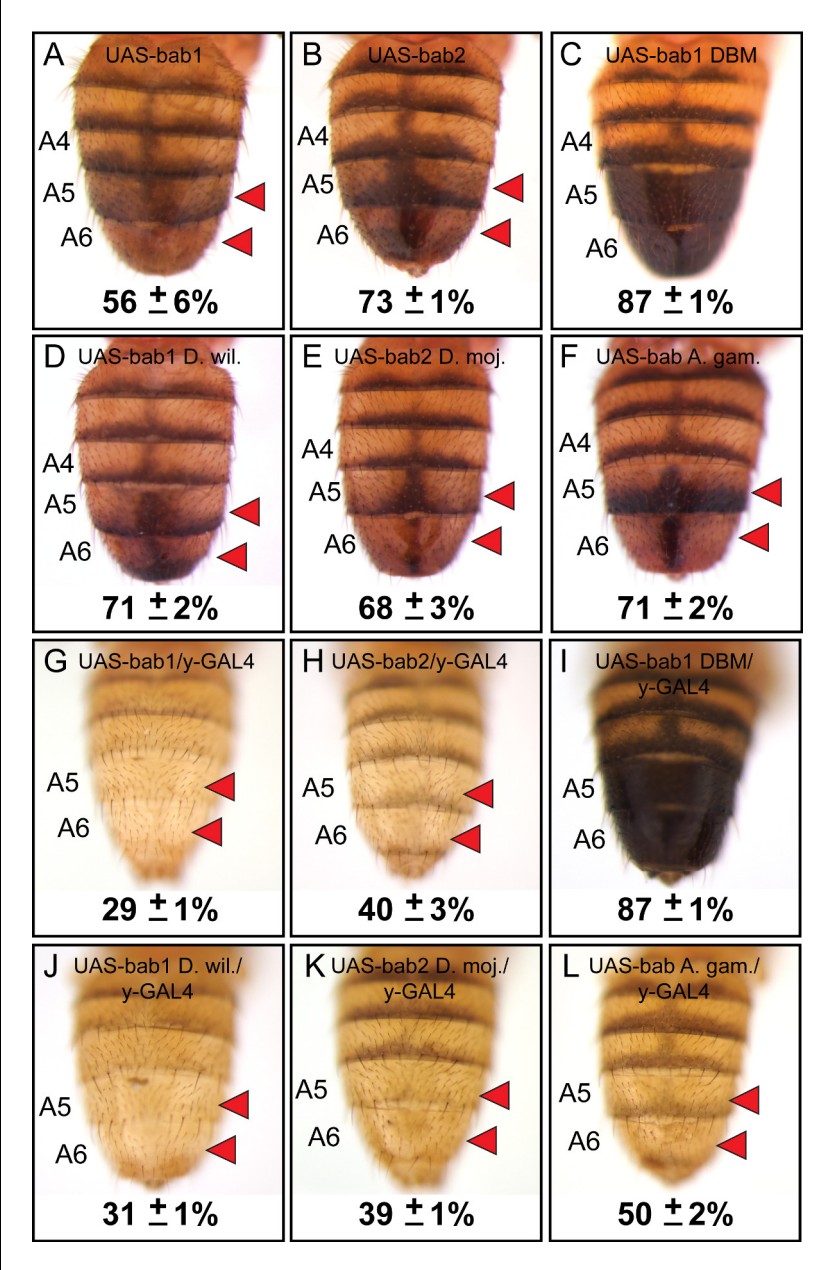

**Figure 6.** Bab1 and Bab2 are sufficient to suppress tergite pigmentation as DNA-binding transcription factors. (A–L) Ectopic expression assays for various *bab* protein coding sequences. (A and G) *D. melanogaster bab1*, (B and H) *D. melanogaster bab2*, (C and I) a DNA-binding compromised version of *D. melanogaster bab1* (bab1 DBM), (D and J) *D. willistoni bab1*, (E and K) *D. mojavensis bab2*, and (F and L) *A. gambiae bab*. (A–F) Leaky expression of transgenes from the attP40 transgene insertion site. (G–L) Ectopic expression of protein coding sequences in the male abdomen under the control of the *y*-GAL4 transgene. Red arrowheads indicate tergite regions with conspicuously reduced tergite pigmentation. The A5 and A6 tergite regions were quantified for their darkness percentage for replicate specimens (n = 4). These percentages and their standard error of the mean (±SEM) are provided below a representative image.

DOI: https://doi.org/10.7554/eLife.32273.013

The following source data and figure supplements are available for figure 6:

**Source data 1.** The DNA and translated protein sequences for the *bab* open-reading frames.
DOI: https://doi.org/10.7554/eLife.32273.018

**Source data 2.** Analysis of pigmentation phenotypes from bab open-reading frame ectopic expression.
DOI: https://doi.org/10.7554/eLife.32273.019

*Figure 6 continued on next page*

*Figure 6 continued*

**Figure supplement 1.** Lethality from ectopic expression of orthologous *bab* open reading frame transgenes in the *pnr* pattern.
DOI: https://doi.org/10.7554/eLife.32273.014

**Figure supplement 2.** Bab proteins are sufficient to suppress tergite pigmentation when ectopically expressed in the dorsal midline of *D. melanogaster*.
DOI: https://doi.org/10.7554/eLife.32273.015

**Figure supplement 3.** The temporal and spatial domains of activity for GAL4 drivers in *D. melanogaster* pupa.
DOI: https://doi.org/10.7554/eLife.32273.016

**Figure supplement 4.** Ectopic expression of the Bab1-DNA-binding mutant protein.
DOI: https://doi.org/10.7554/eLife.32273.017

*GAL4*, (*Figure 6—figure supplement 2*) and *y-GAL4* (*Figure 6J D. willistoni bab1*; and 6K, *D. mojavensis bab2*). These results suggest that the ability of the Bab1 and Bab2 proteins to regulate dimorphic pigmentation did not require the evolution of an altogether new biochemical capability.

The origin of the Bab1 and Bab2 paralogs likely resulted from a duplication event that occurred in the evolutionary history of the Dipteran order (*Figure 2A*). This scenario is supported by the findings that all species of fruit flies with sequenced genomes and the Tsetse fly *Glossina morsitans* possess *bab1* and *bab2* paralogs, whereas basally branching Dipteran species such as *Anopheles* (*A.*) *gambiae* and *Aedes aegypti* only possess a single paralog. Moreover, the genomes of the flour beetle *Tribolium castaneum* and the moth *Bombyx mori*, from orders closely related to Diptera, each possess a single *bab* gene. We were curious as to whether the pigmentation and gene regulatory functions of the derived *bab* paralogs were present in the pre-duplication *bab* gene. To test this hypothesis, we created a UAS-regulated ORF transgene for the *A. gambiae bab* gene to use as a surrogate for the pre-duplication ancestral *bab* gene. We found that like all other *bab* orthologs tested in this study, leaky expression of the *A. gambiae bab* gene resulted in reduced pigmentation in the male A5 and A6 tergites (*Figure 6F*). Moreover, forced ectopic expression driven by *pnr-GAL4* and *y-GAL4* both resulted in more extensive repression of tergite pigmentation (*Figure 6—figure supplement 2*, and *Figure 6L*). These outcomes are consistent with the interpretation that the general functional capability of these proteins was ancient and has been conserved during the 100 million years or more since the gene duplication event occurred. However, these experiments cannot rule out the possibility that some changes have occurred in the *bab* protein coding sequences that cause incremental differences in the ability to regulate dimorphic pigmentation. In fact, the further reduced pigmentation caused by the Bab1 orthologs compared to those for Bab2, and *A. gambiae* Bab suggests that such incremental enhancements have indeed occurred (*Figure 6*).

We sought to determine whether the functional equivalence of the distantly related Bab orthologs can also be seen at the level of target gene regulation. Here, we focused on the capability of the orthologs to repress the expression of the Enhanced Green Fluorescent Protein (EGFP) reporter transgene expression driven by the sequence 5' of *yellow* exon one that contains the wing element and body element CREs (*Figure 7A*) from the dimorphic species *D. melanogaster* (*Figure 7B–G*) and *D. malerkotliana* (*Figure 7H–M*). Orthologs were ectopically expressed in the pupal abdomens of *D. melanogaster* under the control of the *y-GAL4* transgene and the intensity of EGFP expression in the A5 and A6 segment epidermis relative to that for specimens ectopically expressing the Bab1 DBM was measured (*Figure 7B and H*). We observed the *Drosophila* and *Anopheles* orthologs were similarly capable of repressing reporter transgenes regulated by the *yellow* regulatory regions of both dimorphic species. Thus, it can be concluded that the functional equivalency of distantly related Bab orthologs extends to the ability to regulate a derived target gene in a *Drosophila* lineage.

## The gain of direct Bab regulation required CRE evolution for *yellow*

The male-specific pigmentation of the A5 and A6 tergites is thought to be a derived state in the lineage of *D. melanogaster* after it diverged from monomorphic lineages such as that of the *willistoni* and possibly the *obscura* species groups (Figure 9). This may have occurred by two distinct possible routes. First, the regulation of *yellow* expression by Bab might predate the dimorphic pattern of tergite pigmentation, and thus, when Bab expression evolved dimorphism, the *yellow* gene became restricted to males. Alternately, the regulation of *yellow* by Bab may have originated

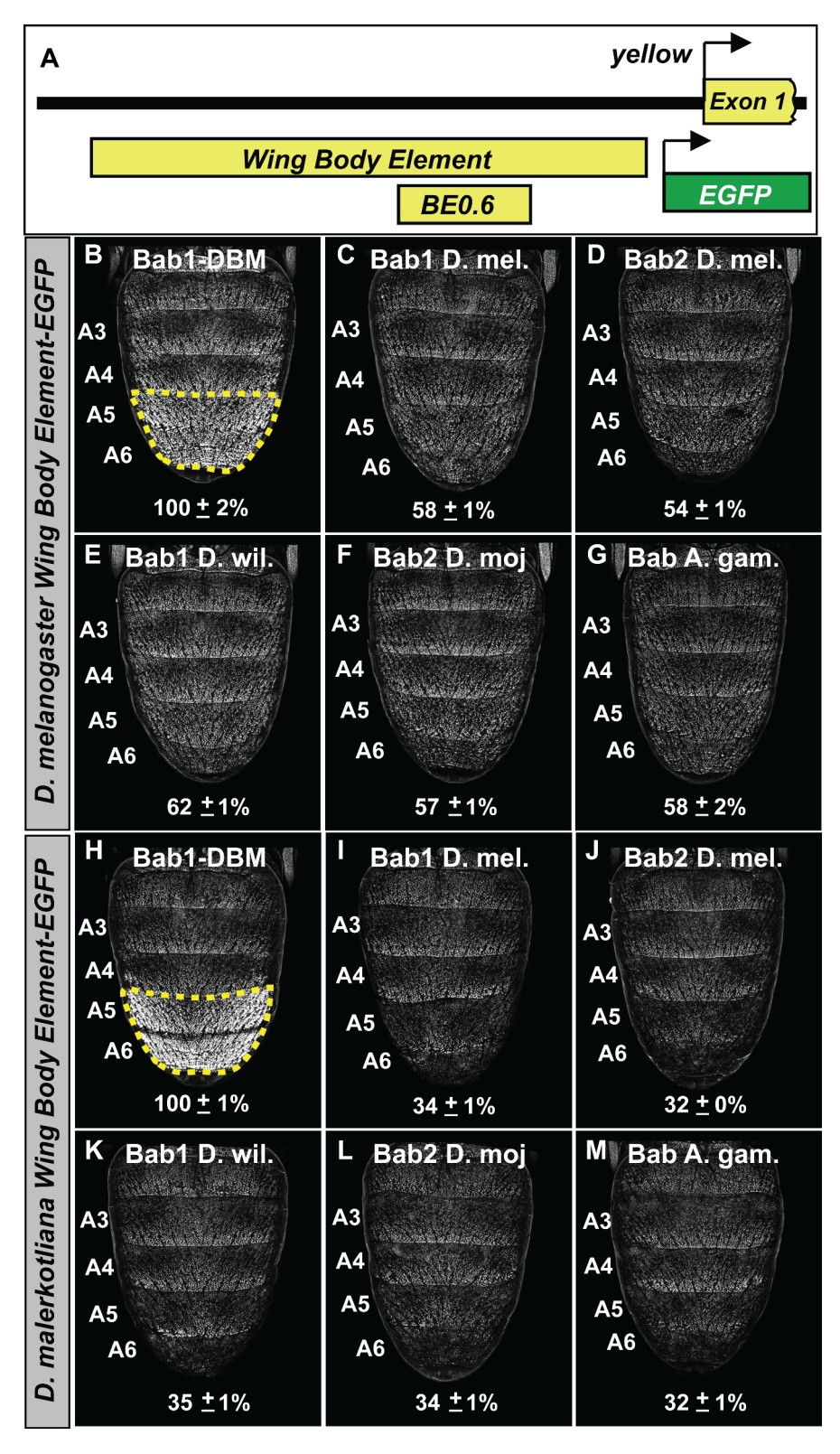

**Figure 7.** The Bab paralogs can suppress the male-specific activity of the regulatory region containing the wing element and body element CREs. (A) 5' of *yellow* exon one resides the wing element and body element CREs, and the position of the *D. melanogaster* yBE0.6 is shown below the to-scale representation of the partial locus. (B–G) Comparison of the levels of EGFP-reporter expression in the male A5 and A6 segments driven by the Wing Body Element of *D. melanogaster*. (H–M) Comparison of the levels of EGFP-reporter expression in the male A5 and A6 segments driven by the Wing

*Figure 7 continued on next page*

*Figure 7 continued*

Body Element of *D. malerkotliana*. The levels of EGFP expression are represented as the % of the mean ±SEM for samples in which the Bab1-DBM was expressed. (**B and H**) Robust EGFP reporter expression in samples ectopically expressing the Bab1-DBM protein in the *y*-GAL4 pattern. (**C and I**) Ectopic expression of Bab1 in the *y*-GAL4 pattern reduced A5 and A6 expression compared to the control. (**D and J**) Ectopic expression of Bab2 in the *y*-GAL4 pattern reduced A5 and A6 expression compared to the control. (**E and K**) Ectopic expression of *D. willistoni* Bab1 in the *y*-GAL4 pattern reduced A5 and A6 expression compared to the control. (**F and L**) Ectopic expression of *D. mojavensis* Bab2 in the *y*-GAL4 pattern reduced A5 and A6 expression compared to the control. (**G and M**) Ectopic expression of *A. gambiae* Bab in the *y*-GAL4 pattern reduced A5 and A6 expression compared to the control.

DOI: https://doi.org/10.7554/eLife.32273.020

The following source data is available for figure 7:

**Source data 1.** Analysis of ectopic expression effects of bab open-reading frames on the dimorphic activities of yellow gene CREs.
DOI: https://doi.org/10.7554/eLife.32273.021

contemporaneously with the evolution of dimorphic Bab expression. We compared the *D. melanogaster* Bab-binding sites from the SM4 and SM10 regions to the orthologous gene regions from species descending from either a dimorphic or monomorphic pigmented ancestor (Figure 9A). This analysis revealed orthologous sequences with several conserved nucleotides among the body elements of species descending from a dimorphic pigmented common ancestor (Figure 9A, node 1). This pattern of sequence conservation is similar to what was found for the derived binding sites for the Hox transcription factor Abd-B that were shown to be a key event in the evolution of male-limited *yellow* expression (*Jeong et al., 2006*). However, these orthologous sequences possess numerous divergent nucleotides among species that share a dimorphic pigmented ancestor (Figure 9A, node 1), and a near complete absence of sequence conservation with species possessing the ancestral monomorphic trait. Thus, it is possible that an ancestral regulatory linkage between Bab and *yellow* is obscured by the turnover and displacement (*Hare et al., 2008*; *Ludwig et al., 2000*; *Swanson et al., 2011*) of Bab-binding sites to other regions of the body element.

To infer the antiquity of Bab-repression at *yellow*, we compared the capabilities of the *D. melanogaster* Bab orthologs to affect the CRE activities of *yellow* 5′ regulatory regions (containing both the wing and body elements) from dimorphic species (*D. melanogaster* and *D. malerkotliana*) and ancestrally monomorphic species (*D. pseudoobscura* and *D. willistoni*) (*Figure 8*). For regulatory sequences derived from dimorphic species, reporter expression in A5 and A6 segments was strikingly reduced in the presence of either ectopic Bab1 or Bab2 compared to the Bab1 DBM control (*Figure 8A–A′′ and and B–B′′*). In contrast, the regulatory sequences from monomorphic species showed modest and no apparent reduction of *D. pseudoobscura* and *D. willistoni* CREs, respectively. Two additional observations can be made. One being the even greater level of regulatory-activity repression for the *D. malerkotliana yellow* regulatory region (33 ± 0% and 33 ± 1% in the presence of ectopic Bab1 or Bab2, respectively) than for *D. melanogaster* (56 ± 1% and 53 ± 0% in the presence of ectopic Bab1 and Bab2, respectively). Second, the regulatory activity of the *D. pseudoobscura* CRE was modestly repressed in the presence of Bab (85 ± 2% and 70 ± 1% in the presence of ectopic Bab1 and Bab2, respectively).

While A5 and A6 *yellow* expression is largely governed by the body element in males, the wing element does direct moderate levels of expression throughout the abdomen and strong expression in posterior stripes of each segment. To see whether the wing element might also respond to Bab expression and thus harbor Bab-binding sites, we compared the levels of expression for the reporter transgenes in the A3 segment in which expression is driven exclusively by the wing element (*Figure 8—figure supplement 1*). While the *D. melanogaster* wing element showed a slight reduction in activity when in the presence of ectopic Bab1 or Bab2 compared to the Bab1 DBM control, no apparent reduction in activity was observed for the transgenes with either the *D. malerkotliana*, *D. pseudoobscura*, or *D. willistoni yellow* regulatory sequences (*Figure 8—figure supplement 1*). These results can be explained by two evolutionary scenarios. First, the linkage between Bab and *yellow* could have been present in a common ancestor of *D. melanogaster*, *D. pseudoobscura*, and *D. willistoni* and was subsequently reduced and lost in the latter two species' lineages, respectively. On the other hand, direct regulation of *yellow* by Bab evolved specifically within the body element, coincident with the evolution of the dimorphic pigmentation trait. This second scenario is consistent with the favored model of *Sophophora* pigmentation evolution (*Jeong et al., 2006*; *Rebeiz and*

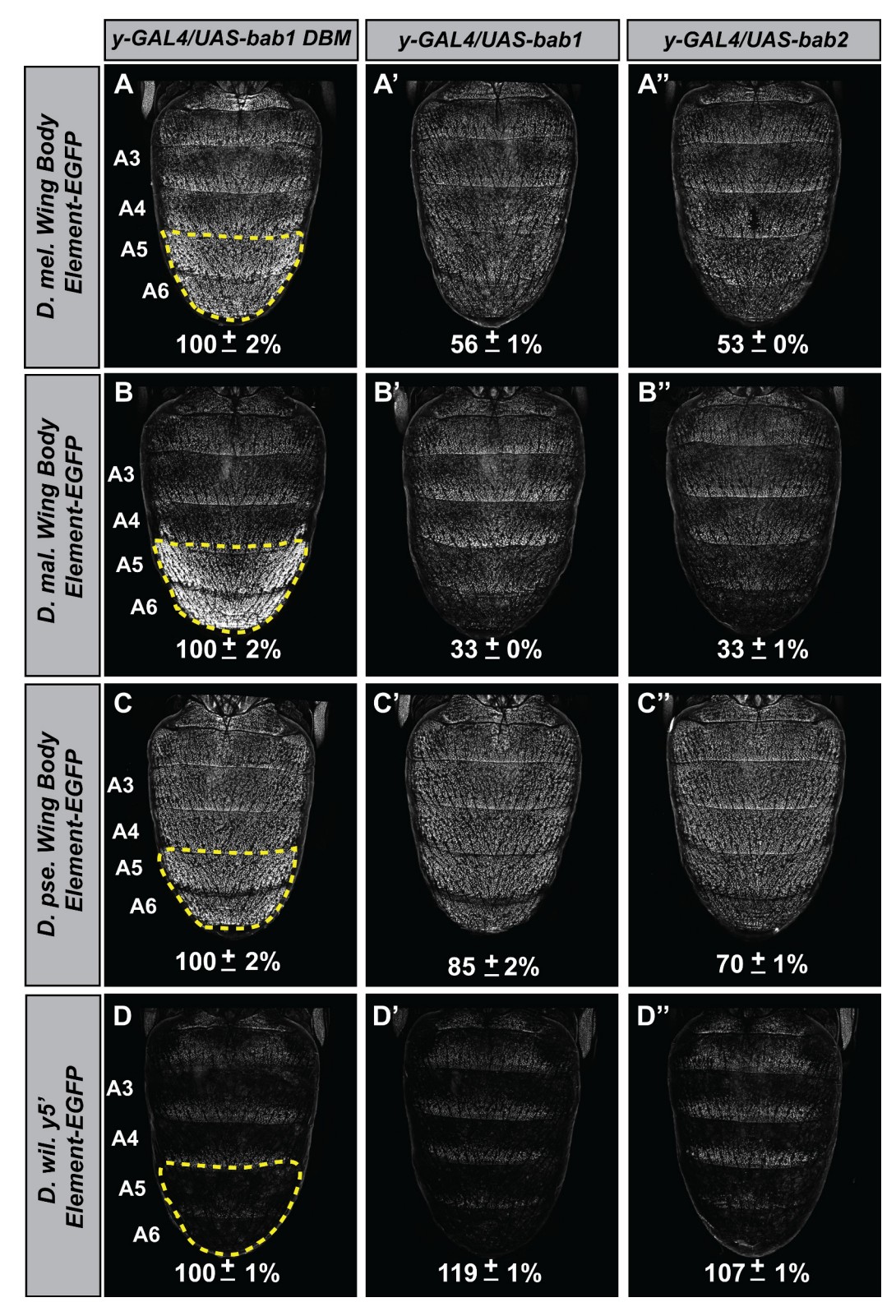

**Figure 8.** The evolved repression by Bab for *cis*-regulatory regions 5′ of *yellow*. (**A–A′′**) Comparison of the levels of EGFP-reporter expression in the male A5 and A6 segments driven by the Wing Body Element of *D. melanogaster*. (**B–B′′**) Comparison of the levels of EGFP-reporter expression in the male A5 and A6 segments driven by the Wing Body Element of *D. malerkotliana*. (**C–C′′**) Comparison of the levels of EGFP-reporter expression in the male A5 and A6 segments driven by the Wing Body Element of *D. pseudoobscura*. (**D–D′′**) Comparison of the levels of EGFP-reporter expression in the

*Figure 8 continued on next page*

*Figure 8 continued*

male A5 and A6 segments driven by the 5′ non-coding region of *D. willistoni yellow*. For each comparison, the level of EGFP expression are expressed as the percentage of the mean ±SEM for samples in which the Bab1-DBM was expressed. (**A–D**) Ectopic expression of the Bab1-DBM in the *y*-GAL4 pattern. (**A′–D′**) Ectopic expression of Bab1 in the *y*-GAL4 pattern. (**A″–D″**) Ectopic expression of Bab2 in the *y*-GAL4 pattern.

DOI: https://doi.org/10.7554/eLife.32273.022

The following source data and figure supplement are available for figure 8:

**Source data 1.** Analysis of the ectopic expression effects of bab open reading frames on yellow gene CREs from species with dimorphic and monomorphic tergite pigmentation.

DOI: https://doi.org/10.7554/eLife.32273.024

**Figure supplement 1.** Orthologous regulatory regions 5′ of the *yellow* gene differ in their responsiveness to *bab*.

DOI: https://doi.org/10.7554/eLife.32273.023

*Williams, 2017*). In the future, deeper taxon sampling in *Sophophora* may shed light on which of the two scenarios reflects the true evolutionary history for this dimorphic pigmentation trait.

## Discussion

Here, we investigated the functions and evolution of the paralogous *D. melanogaster* Bab1 and Bab2 proteins that perform a key regulatory role in a GRN controlling a male-specific pigmentation trait. Although these two paralogs descend from an ancient duplication event (*Figure 2A*), our results show that their ability to function in *D. melanogaster* pigmentation required little-to-no alteration in the functional capability of the Bab proteins. Rather, our data point to the evolution of binding sites in a CRE of a key pigmentation gene, *yellow*, to which these proteins bind through their DNA binding domains (*Figure 9A*). These conclusions are supported by multiple lines of evidence: (1) identification of regions in the *yellow* body CRE that mediate sexual dimorphism, (2) in vitro binding of Bab proteins to these sequences, (3) the ability of pre-duplicate Bab proteins to suppress the *yellow* body CRE of dimorphic species, and (4) the relative inability of these proteins to exert similar effects on CREs of species whose lineages predate the evolution of this trait. These findings represent the first-documented direct target sites of the Bab proteins, sites which arose coincident with the evolution of sexually dimorphic Bab expression patterns in the abdomen (*Figure 9B*). Thus, our results provide a clear example in which multiple tiers of a complex GRN evolved to produce a Hox-regulated trait while preserving the genetic toolkit of regulatory and differentiation genes.

### The evolution of the Bab paralogs

The *D. melanogaster bab* locus provides an interesting example in which the protein-coding and regulatory divergence of duplicated genes can be compared. The phylogenetic distribution of *bab* paralogs supports an estimated timing of the duplication event to around 125 million years since the common ancestor of the fruit fly, Tsetse fly, and Hessian fly split from the lineage of the sandfly and mosquitoes (*Figure 2A*) (*Wiegmann et al., 2011*). Duplicate genes may sub-functionalize, neo-functionalize, or be lost through pseudogenization (*Lynch and Conery, 2000*). Since this duplication event, both *bab* paralogs have been maintained in the genomes of distantly related fruit fly species (*Clark et al., 2007*; *Richards et al., 2005*), and in species from related families whose genomes have been sequenced (*Giraldo-Calderón et al., 2015*; *Kriventseva et al., 2015*). Here, we showed that both *D. melanogaster bab* paralogs and orthologs from other fly and mosquito lineages can similarly impact the development of melanic tergite pigmentation when ectopically expressed in *D. melanogaster* (*Figure 6*), and each gene can induce a similar non-specific split tergite phenotype (*Figure 6—figure supplement 2*). These outcomes suggest for at least for body pigment regulation, that their protein coding regions are functionally equivalent.

In contrast to the conserved protein functionality that we observed for the Bab paralogs, some divergent patterns of expression have been found for the paralogs in *D. melanogaster,* consistent with a role for neo- or sub-functionalization (*Couderc et al., 2002*). Yet, the majority of *bab* paralog expression patterns appear to be common to both paralogs (*Couderc et al., 2002*; *Rogers et al., 2013*; *Salomone et al., 2013*), including the pupal abdominal epidermis, for which expression is governed by two shared CREs (*Williams et al., 2008*). It has been found that heterozygous *bab* null females express a male-like pattern of tergite pigmentation compared to wild type females, and the

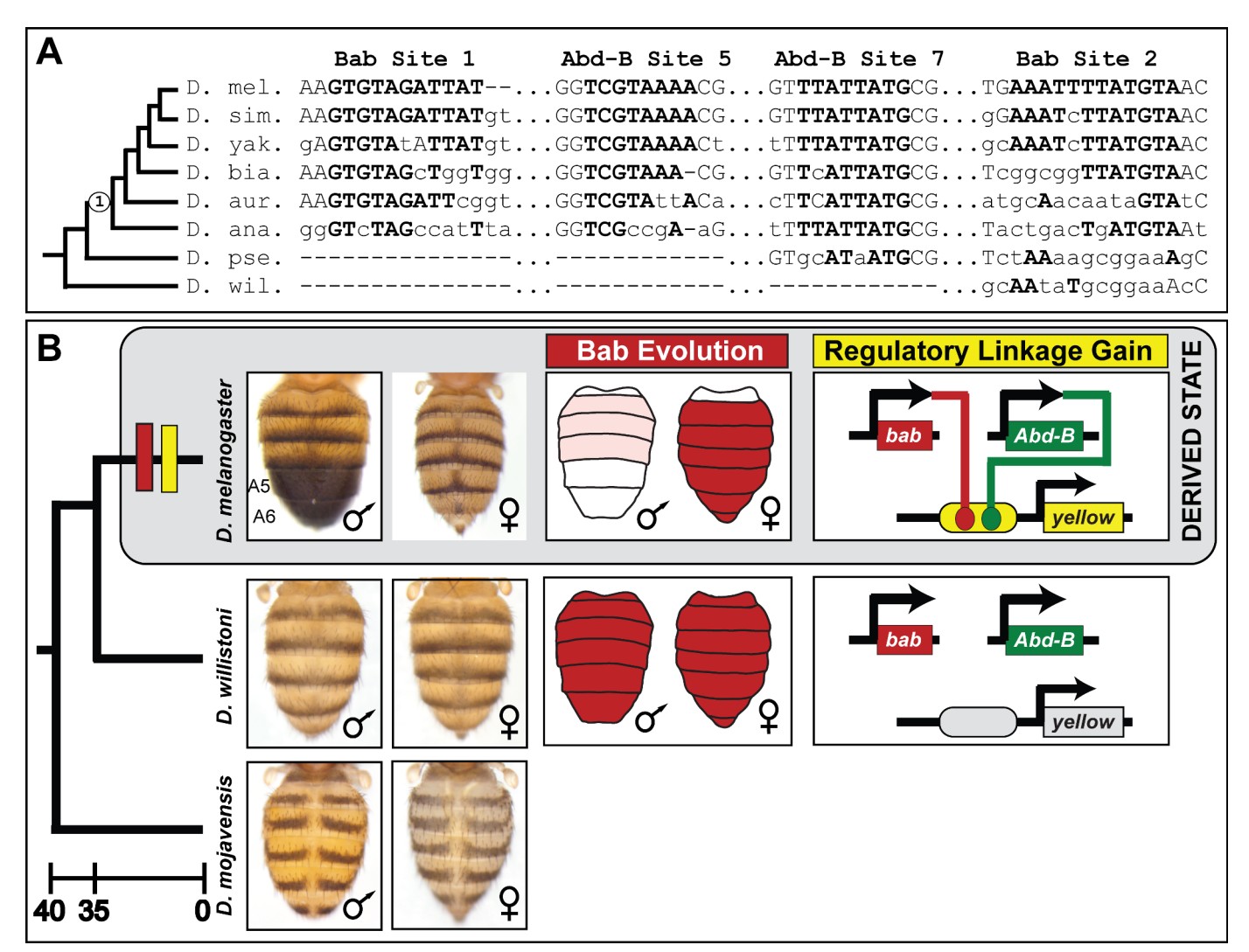

**Figure 9.** The evolution of male-specific pigmentation required the gain of a regulatory linkage between Bab and the newly evolved body element CRE controlling *yellow* expression. (**A**) An alignment of the Bab-bound sequences in the SM4 (site 1) and SM10 (site 2) regions and for the two previously identified binding sites for Abd-B in the yBE0.6 CRE (*Jeong et al., 2006*). 'Node 1' on the phylogeny indicates the most recent common ancestor suspected to have possessed the derived male-specific pattern of pigmentation. Time scale shown is in millions of years ago. Bold capital letters indicate the bases bound by the transcription factor in the *D. melanogaster* CRE, and those which are conserved in the orthologous regions for related species. (**B**) Model for the derivation of a dimorphic pigmentation trait where dimorphic pigmentation required the evolution of a dimorphic Bab expression and the gain of a regulatory linkage between Bab and *yellow* through gains of binding sites in the body element CRE.
DOI: https://doi.org/10.7554/eLife.32273.025

homozygous null pigmentation phenotype is more or less equivalent to that of males. (*Rogers et al., 2013*). Here, we showed that the RNAi reduction of expression for either *bab1* or *bab2* resulted in more male-like pigmentation pattern on the female abdomen, and RNAi for both paralogs simultaneously resulted in a more pronounced male-like phenotype (*Figure 5*). While qualitative differences through CRE functional divergence must have occurred to drive the divergent paralog expression in some tissues, the need for a higher overall quantity of expressed Bab protein seems to be key for Bab's role in the GRN generating the derived dimorphic pattern of abdomen pigmentation.

## Bab and its history in a pigmentation gene regulatory network

The stark dimorphism between the melanic pigmentation of *D. melanogaster* male and female abdominal tergites represents a trait whose origin has now been resolved to the level of its GRN connections. Dimorphic pigmentation is thought to have derived from a monomorphic ancestral state in the lineage of *D. melanogaster* after it diverged from that for *D. willistoni* and perhaps even as recently as the *D. pseudoobscura* split in the *Sophophora* subgenus (*Jeong et al., 2006*; *Salomone et al., 2013*). Assuming this scenario is generally correct, then when and how did Bab become a part of this GRN? One plausible explanation is that Bab regulated the expression of pigmentation genes prior to the emergence of this dimorphic trait, perhaps as a part of an antecedent dimorphic GRN. One study provided data consistent with this scenario, showing that Bab2 expression often, but not always, displayed an anti-correlation to where melanic pigmentation developed on fruit fly tergites, including non-*Sophophora* species (*Gompel and Carroll, 2003*). If Bab had an ancient role in regulating the expression of pigmentation genes such as *yellow*, then dimorphic pigmentation could have evolved by re-deploying a conserved Bab-responsive CRE in the abdomen. Furthermore, it was demonstrated that an abdominal epidermis CRE activity is present in the *yellow* gene intron of the monomorphic pigmented species such as *D. pseudoobscura*, *D. willistoni*, and the non-Sophophoran species *D. virilis* (*Kalay and Wittkopp, 2010*). Thus, it is conceivable for the Bab-body element linkage to have been present, while the monomorphic activity of this second *yellow* CRE masked the dimorphic regulation of the body element.

However, various data are difficult to reconcile with this re-deployment model. First, the melanic species *D. virilis* expresses Bab1 and Bab2 throughout the male and female abdomens suggesting an inability of Bab to suppress pigmentation genes such as *yellow* in this species (*Salomone et al., 2013*). Furthermore, the regulation of *yellow* by Bab in *D. melanogaster* is limited to the body element CRE in which we found Bab binding sites. This is significant as clearly identifiable orthologs to the *D. melanogaster* body element remains difficult to identify in more distantly related fruit fly species hailing from monomorphically pigmented lineages (*Figure 9A*), although the wing element and its activity has remained well-conserved by comparison. Thus, there is no evidence for an ancient linkage between Bab and an antecedent of the *yellow* body element. While an ancestral *bab-yellow* linkage may exist for a yet unidentified trait, the activation of *yellow* in the well-studied male-specific spot that adorns the wings of *D. biarmipes* is *bab*-independent (*Arnoult et al., 2013*; *Gompel et al., 2005*). Finally, the *D. melanogaster* body element possesses derived binding sites for both Bab and the Hox factor Abd-B (*Figure 9A*) (*Jeong et al., 2006*). Thus, the co-option of either Bab or Abd-B would be insufficient to account for the origin of this dimorphic trait.

Based upon the available data, it seems much more likely that Bab was integrated into an antecedent GRN to play a key role in differentiating the expression outcomes between males and females. This integration involved the remodeling of existing CREs controlling Bab expression in the abdominal epidermis (*Rogers et al., 2013*; *Williams et al., 2008*), and the acquisition of *yellow* as a direct target of regulation through the formation of binding sites in the emergent body element (*Figure 9*). At this point, it is unclear whether dimorphic Bab expression preceded the evolution of Bab sites in the *yellow* body element CRE or if the Bab sites evolved first. One hint to this puzzle may lie in *D. pseudoobscura,* a species whose *yellow* body element CRE is mildly Bab-responsive (*Figure 8* and *Figure 8—figure supplement 1*), but for which Bab expression retains its ancestral monomorphic expression. This might suggest that the capability to respond (albeit weakly) to Bab evolved first, followed by the evolution of dimorphic Bab expression patterns. Further, the evolutionary connections identified here represent a subset of the connections downstream of Bab. While the *yellow* body CRE contains separable activating and repressing inputs, the gene *tan*, which is co-expressed with *yellow* appears to have a very different encoding for dimorphism (*Camino et al., 2015*). Extensive mutagenesis of the *tan* MSE failed to find any mutations that relaxed dimorphic expression, suggesting that activating and repressing inputs are overlapping, or are closely situated in this CREs DNA sequence. Future studies of this GRN will illuminate how the network downstream of Bab was elaborated.

## The incorporation of old transcription factors into new networks

We suggest that the increased complexity of the dimorphic pigmentation GRN through the integration of the Bab transcription factors by CRE evolution will exemplify a common mechanism whereby

increasingly sophisticated GRNs have come about to regulate traits throughout the animal kingdom. The vast majority of transcription factor binding specificities remain conserved over long evolutionary periods (*Nitta et al., 2015*), and many of these factors are functionally equivalent between distantly related taxa. The exceptions to this trend may represent rare examples of transcription factor diversification that occurred in the distant past, and thus are limited to a vanishingly small number of traits, or may represent examples of developmental systems drift in which the molecular mechanisms change, but the outcome remains the same (*True and Haag, 2001*). Studying more recent trait divergence allows one to more clearly discern phenotypically relevant evolutionary changes from those involving systems drift. Tests of regulatory sequence divergence are particularly hampered by drift, as sequence divergence is rapid and CREs from distantly related taxa often work poorly in heterologous transgenic environments. Thus, further comparisons of genetically tractable traits that arose over similarly recent timescales, in which protein coding and *cis*-regulatory divergence can be directly compared in vivo are required to unveil the nature of this broader trend.

# Materials and methods

## Key resources table

| Reagent type (species) or resource | Designation | Source or reference | Identifiers |
|---|---|---|---|
| Gene (*Drosophila melanogaster*) | bab1 | | FLYB:FBgn0004870 |
| Gene (*D. melanogaster*) | bab2 | | FLYB:FBgn0025525 |
| Gene (*D. melanogaster*) | yellow | | FLYB:FBgn0004034 |
| Genetic reagent (*D. melanogaster*) | pnr-Gal4 | Bloomington Drosophila Stock Center | BDSC:3039 |
| Genetic reagent (*D. melanogaster*) | y-Gal4 | Bloomington Drosophila Stock Center | BDSC:44267; FLYB:FBst0044267; RRID:BDSC_44267 |
| genetic reagent (*D. melanogaster*) | UAS-GFP-nls | Bloomington Drosophila Stock Center | BDSC:4776 |
| Genetic reagent (*D. melanogaster*) | UAS-mCherry dsRNA | Bloomington Drosophila Stock Center | BDSC:35785 |
| Genetic reagent (*D. melanogaster*) | UAS-bab1 ORF of D. melanogaster | this paper | |
| Genetic reagent (*D. melanogaster*) | UAS-bab2 ORF of D. melanogaster | this paper | |
| Genetic reagent (*D. melanogaster*) | UAS-bab1 DBM ORF of D. melanogaster | this paper | |
| Genetic reagent (*D. melanogaster*) | UAS-bab1 ORF of D. willistoni | this paper | |
| Genetic reagent (*D. melanogaster*) | UAS-bab2 ORF of D. mojavensis | this paper | |
| Genetic reagent (*D. melanogaster*) | UAS-bab Anopheles gambiae | this paper | |
| Genetic reagent (*D. melanogaster*) | UAS-bab1 siRNA id #3 | this paper | |
| Genetic reagent (*D. melanogaster*) | UAS-bab1 siRNA id #4 | this paper | |
| Genetic reagent (*D. melanogaster*) | UAS-bab2 siRNA id #12 | this paper | |
| Genetic reagent (*D. melanogaster*) | UAS-bab2 siRNA id #16 | this paper | |
| Genetic reagent (*D. melanogaster*) | UAS-bab1 siRNA id #3 + bab2 siRNA id#12 | this paper | |
| Genetic reagent (*D. melanogaster*) | UAS-bab1 siRNA id #3 + bab2 siRNA id#16 | this paper | |
| Genetic reagent (*D. melanogaster*) | yBE0.6-EGFP reporter | PMID:25835988 | |
| Genetic reagent (*D. melanogaster*) | yBE0.6 SM4-EGFP reporter | PMID:25835988 | |
| Genetic reagent (*D. melanogaster*) | yBE0.6 SM10-EGFP reporter | PMID:25835988 | |
| Genetic reagent (*D. melanogaster*) | yBE0.6 SM4 + SM10 EGFP reporter | PMID:25835988 | |
| Genetic reagent (*D. melanogaster*) | yWing + Body Element (D. melanogaster)-EGFP reporter | PMID:25835988 | |

*Continued on next page*

*Continued*

| Reagent type (species) or resource | Designation | Source or reference | Identifiers |
|---|---|---|---|
| Genetic reagent (*D. melanogaster*) | yWing + Body Element (D. malerkotliana)-EGFP reporter | PMID:25835988 | |
| Genetic reagent (*D. melanogaster*) | yWing + Body Element (D. pseudoobscura)-EGFP reporter | PMID:25835988 | |
| Genetic reagent (*D. melanogaster*) | y5'1 (D. willistoni)-EGFP reporter | PMID:25835988 | |
| Antibody | anti-Bab1 (rabbit polyclonal) | PMID:18724934 | |
| Antibody | Alexa 647-secondary | Invitrogen | A-21244 |
| Recombinant DNA reagent | GST-Bab1 DNA Binding Domain (DBD) | this paper | |

## Fly stocks and genetic crosses

All fly stocks were cultured at 22°C using a sugar food medium (*Salomone et al., 2013*). The yBE0.6, yBE0.6 SM4, yBE0.6 SM10, and SM4 +10 reporter transgenes utilized in GAL4/UAS experiments were each inserted into the attP40 site (*Camino et al., 2015*; *Markstein et al., 2008*). GAL4 expression was driven in the pattern of the *pannier* (*pnr*) gene using the *pnr*-GAL4 chromosome (*Calleja et al., 2000*) and the pupal abdominal epidermis pattern (*Jeong et al., 2006*; *Wittkopp et al., 2002*) of the *yellow* gene using the *y*-GAL4 transgene (*Hart, 2013*). The *pnr*-GAL4 (BDSC ID#3039) and *y*-GAL4 (BDSC ID#44267) fly stocks were obtained from the Bloomington Drosophila Stock Center. A UAS-mCherry dsRNA line (BDSC ID#35785) was used as a negative control in the RNA-interference experiments. The reporter transgenes containing orthologous sequences 5' of the *yellow* first exon adjacent to a minimal *hsp70* promoter and the coding sequence of the EGFP-NLS reporter protein were integrated into the attP2 site on the *D. melanogaster* third chromosome whose construction was previously described (*Camino et al., 2015*; *Groth et al., 2004*).

## Recombinant protein production and gel shift assays

The protein coding sequence for amino acids 490–672 of the *D. melanogaster* Bab1protein was cloned into the *BamHI* and *NotI* sites of the pGEX4T-1 vector in order to express an N-terminal GST-fusion protein that has the AT-Hook and psq domains and that possesses DNA binding capability (*Lours et al., 2003*). This vector was transformed into the BL21 DE3 *E. coli* strain (New England Biolabs) and recombinant protein was purified by a standard protocol (*Williams et al., 1995*) with slight modifications. In brief, an overnight bacterial culture was grown at 37°C in LB media with 200 µg/ml Ampicillin. This culture was added to 225 ml of a rich LB media (2% Tryptone, 1% Yeast Extract, and 1% sodium chloride) and grown at 37°C. After 1 hour (hr) of growth, protein expression was induced by adding IPTG to a final concentration of 0.5 mM, and cultured for an additional 3 hr. Bacteria were then pelleted by centrifugation, media decanted, and bacterial pellets frozen at −74°C. Bacteria pellets were thawed on ice and resuspended in ice cold STE buffer containing protease inhibitors (Thermo Scientific, Waltham MA. After a 15 minute (min) incubation on ice, DTT was added to 5 mM and Sarkosyl to 1.5%. The bacterial suspension was subjected to 4 rounds of sonication on ice at 33 amps for 1 min each round with, and a 1-min rest between rounds. 1 ml of glutathione agarose (Thermo Scientific) was then added to the suspension and allowed to mix for 15 min with nutation at 4°C. The glutathione-agarose was then washed seven times with ice cold PBS. GST-Bab1 DNA-binding domain (DBD) fusion protein was eluted from the glutathione-agarose by seven 1.5 ml aliquots of protein elution buffer (75 mM Hepes pH 7.4, 150 mM NaCl, 20 mM reduced glutathione, 5 mM DTT, and 0.1% Triton X-100). Collected aliquots were combined and concentrated using Vivaspin 20 spin columns with a 100,000 MWCO (Sartorious). The purified GST-Bab1 DBD protein was snap frozen using a dry ice ethanol bath, and stored in aliquots at −74°C.

Reverse complementary oligonucleotides were synthesized (Integrated DNA Technologies) that contain wild type or mutant yBE0.6 sequences (*Tables 3* and *4*). Gel shift assays were done as previously described (*Camino et al., 2015*; *Rogers et al., 2013*). In brief, all oligonucleotides were biotin-labeled on their 3' end using the DNA 3' End Biotinylation Kit (Thermo Scientific, Waltham MA) using the manufacture's protocol. Biotin-labeled complementary oligonucleotides were annealed to form double stranded probes, and labeling efficiency was evaluated by the manufacturer's

**Table 3.** Oligonucleotides used to make Scan Mutant four region gel shift probes.

| Probe | Sequence (5' to 3') | Oligo name |
|---|---|---|
| SM4 Region 1 | ATTCTTTAATTTGTATTTTAATATT | yBE 4i1 Top |
| | AATATTAAAATACAAATTAAAGAAT | yBE 4i1 Bottom |
| SM4 Region 2 | ATATTTTGAGAGGTTTTCCTTATTTAAAGT | yBE 4i2 Top |
| | ACTTTAAATAAGGAAAACCTCTCAAAATAT | yBE 4i2 Bottom |
| SM4 Region 3 | AAAGTGTAGATTATTGAGGATTAAT | yBE 4i3 Top |
| | ATTAATCCTCAATAATCTACACTTT | yBE 4i3 Bottom |
| SM4 Region 3 Scan Mutant | cAcGgGgAtAgTcTgGcGtAgTcAg | y4i3 T Scrm |
| | cTgAcTaCgCcAgAcTaTcCcCgTg | y4i3 B Scrm |
| Region 3 TA > GA | AAAGTGgAGATgATTGAGGATgAAT | yBE 4i3 TA > GA Top |
| | ATTcATCCTCAATcATCTcCACTTT | yBE 4i3 TA > GA Bottom |
| SM4 Region 3 sub1 | gggCgggCgATTATTGAGGATTAAT | y4i3 sub1 T |
| | ATTAATCCTCAATAATcGgggGccc | y4i3 sub1 B |
| SM4 Region 3 sub2 | AAAGTgggCgggCgTGAGGATTAAT | y4i3 sub2 T |
| | ATTAATCCTCAcGcccGcccACTTT | y4i3 sub2 B |
| SM4 Region 3 sub3 | AAAGTGTAGgggCgggCgATTAAT | y4i3 sub3 T |
| | ATTAATcGcccGcccTCTACACTTT | y4i3 sub3 B |
| SM4 Region 3 sub4 | AAAGTGTAGATTATTGgggCgggCg | y4i3 sub4 T |
| | cGcccGcccCAATAATCTACACTTT | y4i3 sub4 B |

DOI: https://doi.org/10.7554/eLife.32273.026

**Table 4.** Oligonucleotides used to make Scan Mutant 10 region gel shift probes.

| Probe | Sequence (5' to 3') | Oligo name |
|---|---|---|
| SM10 Region 1 | TCGTCCCTTTTGAAATTTTATGTAACACTC | yBE 10i1 Top |
| | GAGTGTTACATAAAATTTCAAAAGGGACGA | yBE 10i1 Bottom |
| SM10 Region 2 | CACTCAATTATATTTATGTATATGTATGCT | yBE 10i2 Top |
| | AGCATACATATACATAAATATAATTGAGTG | yBE 10i2 Bottom |
| SM10 Region 3 | ATGCTCAAAATCACCTGCCAATAACCCTGCAGG | yBE 10i3 Top |
| | CCTGCAGGGTTATTGGCAGGTGATTTTGAGCAT | yBE 10i3 Bottom |
| SM10 Region 1 Scan Mutant | gCtTaCaTgTgGcAcTgTgAgGgAcCcCgC | y10i1 T Scrm |
| | GcGgGgTcCcTcAcAgTgCcAcAtGtAaGc | y10i1 B Scrm |
| SM10 Region 3 Scan Mutant | cTtCgCcAcAgCcCaTtCaAcTcAaCaTtCcGt | y10i3 T Scrm |
| | aCgGaAtGtTgAgTtGaAtGgGcTgTgGcGaAg | y10i3 B Scrm |
| SM10 Region 1 sub1 | gggCgggCgggCAAATTTTATGTAACACTC | y10i1 sub1 T |
| | GAGTGTTACATAAAATTTGcccGcccGccc | y10i1 sub1 B |
| SM10 Region 1 sub2 | TCGTCCgggCgggCgggCTATGTAACACTC | y10i1 sub2 T |
| | GAGTGTTACATAGcccGcccGcccGGACGA | y10i1 sub2 B |
| SM10 Region 1 sub3 | TCGTCCCTTTTGgggCgggCgggCACACTC | y10i1 sub3 T |
| | GAGTGTGcccGcccGcccCAAAAGGGACGA | y10i1 sub3 B |
| SM10 Region 1 sub4 | TCGTCCCTTTTGAAATTTgggCgggCgggC | y10i1 sub4 T |
| | GcccGcccGcccAAATTTCAAAAGGGACGA | y10i1 sub4 B |

DOI: https://doi.org/10.7554/eLife.32273.027

quantitative Dot Blot assay. Binding reactions included 25 fmol of probe and GST-Bab1 DBD protein in General Footprint Buffer (working concentration of 50 mM HEPES pH 7.9, 100 mM KCl, 1 mM DTT, 12.5 mM MgCl$_2$, 0.05 mM EDTA, and 17% glycerol). For each probe, separate binding reactions were done that included 4000 ng, 2000 ng, 1000 ng, 500 ng, and 0 ng of the GST-Bab1 DBD protein. Binding reactions were carried out for 30 min at room temperature and then size separated by electrophoresis through a 5% non-denaturing polyacrylamide gel for 60 min at 200 V. Following electrophoresis, the samples were transferred and cross linked to a Hybond-N +membrane (GE Healthcare Amersham) for chemiluminescent detection using the Chemiluminescent Nucleic Acid Detection Module and manufacture's protocol (Thermo Scientific). Chemiluminescent images were recorded using a BioChemi gel documentation system (UVP).

## *bab1* and *bab2* shmiR expressing transgenes

The ORFs for *D. melanogaster* *bab1* and *bab2* were obtained from NCBI accession numbers NM_206229 and NM_079155.3 respectively. From these, nucleotide guide sequences were designed using the Designer of Small Interfering RNA (DSIR) algorithm (*Vert et al., 2006*) that is accessible at: http://biodev.extra.cea.fr/DSIR/DSIR.html. For *bab1*, the eleven rows of output were included in *Table 1*. For *bab2*, there were 54 rows of output, sorted by descending Corrected Score, and the top 19 rows of output presented in *Table 2*. To make sure shmiRs lack the same seed residues (nucleotides 2–8) as those present in known miRNAs, we searched candidate guide sequences against a miRNA database (http://mirbase.org/search.shtml). Search sequences were set to 'Mature miRNAs, E-value cutoff of '10', Maximum hits of 100, and results were shown for 'fly'.

Previously it was shown that a shmiR can induce phenotypes in transgenic flies when the guide shares at least 16–21 base pairs of contiguous sequence to the target gene (*Haley et al., 2010*). Thus, we sought the highest scoring 'Guide' sequences for which fewer than 16 contiguous bases match a heterologous exon sequence in the *D. melanogaster* genome. Guide sequences were evaluated in a BLAST search of the *D. melanogaster* genome (http://flybase.org/blast/) with the word size set to 7. The genomic position of the BLAST hits were identified using the GBrowse feature. An RNAi transgene targeting *bab1* was created by the Transgenic RNAi Project (TRiP) at Harvard Medical School that included the sequence identified here as siRNA_id 1 (*Table 1*) that we have found to be ineffective at suppressing *bab1* expression. Thus, this guide sequence was excluded from further consideration here. For *bab1,* we elected to create shmiRs with the siRNA three and siRNA four sequences which each have a 21 base pair match to a sequence in the *bab1* first exon. For *bab2,* we elected to create shmiRs with the siRNA 16 and siRNA 12 sequences, each which have a 21 base sequence that matches a sequence in the second exon of *bab2*. The *bab1* and *bab2* shmiRs were designed to possess two essential mismatches to maintain a miR-1 stem-loop structure (*Haley et al., 2010*), and oligonucleotides were designed for annealing that have *NheI* and *EcoRI* overhangs for cloning into the pattB-NE3 vector (*Table 5*). The annealed oligonucleotides were cloned into the *NheI* and *EcoRI* sites of the pattB-NE3 vector, and successful cloning was verified by Sanger sequencing using the pUASTR1 primer (5' CCCATTCATCAGTTCCATAGGTTG 3'). pattB-NE3 vectors containing an shmiR guide sequence were site-specifically integrated into the *D. melanogaster* attP2 landing site (*Groth et al., 2004*) by standard protocol (Best Gene Inc.).

## Chaining shmiRs to target *bab1* and *bab2*

shmiR chains were created in two steps. First, the bab1_3 shmiR was removed from the pattB-NE3 vector by HindIII and BamHI digestion and the excised piece was subcloned into the pHB vector (*Haley et al., 2010*, *2008*). The shmiR piece was then amplified from the pHB vector using the M13F and M13R primers. This PCR product was digested with KpnI and SpeI restriction endonucleases and then cloned into the *KpnI* and *XbaI* sites of the pattB-NE3 vectors containing the bab2 siRNA 16 and the vector containing the bab2 siRNA 12 sequence. For each vector, the presence of the tandem shmiR sequences was verified by Sanger sequencing in separate reactions with the PUASTR1 (5' CCCATTCATCAGTTCCATAGGTTG 3') and PUASTF1 (5' ACCAGCAACCAAGTAAATCAACTG3') primers. These chained shmiR transgenes were injected into *D. melanogaster* embryos for site-specific integration into the attP2 site on the third chromosome to make transgenic stocks (*Groth et al., 2004*).

**Table 5.** Oligonucleotides for cloning *bab1* and *bab2* shRNAs into *NheI* and *EcoRI* sites of pattB-NE3 vector.

| siRNA name and sequence | Oligo name | Oligo sequence (5' – 3') |
| --- | --- | --- |
| bab1 siRNA 3 TTCGTCTGATAGTTGTTCCAG | b1_3 Top b1_3 Bottom | ctagcagtCTGGAACAACAATCAGACGTAtagttatattcaagcataTTCGTCTGATAGTTGTTCCAGgcg aattcgcCTGGAACAACTATCAGACGAAtatgcttgaatataactaTACGTCTGATTGTTGTTCCAGactg |
| bab1 siRNA 4 TACAGCATGACCTTGACTCTC | b1_4 Top b1_4 bottom | ctagcagtGAGAGTCAAGCTCATGCTGAAtagttatattcaagcataTACAGCATGACCTTGACTCTCgcg aattcgcGAGAGTCAAGGTCATGCTGTAtatgcttgaatataactaTTCAGCATGAGCTTGACTCTCactg |
| bab2 siRNA 16 TATTTCAAAGTCCACAATCTG | b2_16 Top b2_16 Bottom | ctagcagtCAGATTGTGGTCTTTGAAAAAtagttatattcaagcataTATTTCAAAGTCCACAATCTGgcg aattcgcCAGATTGTGGACTTTGAAATAtatgcttgaatataactaTTTTTCAAAGACCACAATCTGactg |
| bab2 siRNA 12 TTGGACTTCACCAGCTCCGTT | b2_12 Top b2_12 Bottom | ctagcagtAACGGAGCTGCTGAAGTCCTAtagttatattcaagcataTTGGACTTCACCAGCTCCGTTgcg aattcgcAACGGAGCTGGTGAAGTCCAAtatgcttgaatataactaTAGGACTTCAGCAGCTCCGTTactg |
| bab2 siRNA 20 TCGAACTGATCGATTTCGCCG | b2_20 Top b2_20 Bottom | ctagcagtCGGCGAAATCCATCAGTTCCAtagttatattcaagcataTCGAACTGATCGATTTCGCCGgcg aattcgcCGGCGAAATCGATCAGTTCGAtatgcttgaatataactaTGGAACTGATGGATTTCGCCGactg |

DOI: https://doi.org/10.7554/eLife.32273.028

## *Bab* open-reading frame transgenes

The ORFs for *D. melanogaster bab1* and *bab2*, *D. willistoni bab1* (GK16863-PA), *D. mojavensis bab2* (GI12710), and *Anopheles (A.) gambiae bab* (AGAP006018-RA) were customized for gene synthesis by GenScript Incorporated. We added a Syn21 translational enhancer (*Pfeiffer et al., 2012*) 5' of each ORF's initiator ATG, and an additional nonsense codon was added just 3' of the endogenous one. The ORFs were flanked by a 5' *EcoRI* site and a 3' *NotI* site. The *A. gambiae* ORF had the coding sequence for the V5 epitope tag added after the initiator codon. The DNA sequences were modified with synonymous substitutions as needed in order to optimize for gene synthesis. The synthesized sequences can be found in *Figure 6—source data 1* After synthesis, the ORFs were removed from the pUC57 vector backbone and subcloned into the *EcoRI* and *NotI* sites of a pUAST vector modified to possess an attB site (called pUMA) for site-specific transgene integration. All ORFs are under the regulatory control of the vector's upstream activating sequences (UAS sites) to allow for conditional ORF expression by the GAL4/UAS system (*Brand and Perrimon, 1993*).

A *bab1* DNA-binding mutant (DBM) ORF was created that possesses non-synonymous mutations in the Pipsqueak and AT-Hook motifs that results in a protein lacking it's in vitro DNA-binding capability (*Lours et al., 2003*). The coding sequence for the *bab1* Pipsqueak and AT-Hook motifs are flanked by *AscI* and *BamHI* restriction endonuclease sites. We designed a coding sequence within this sequence that includes mutations altering codons with these two protein domains (*Figure 6—source data 1*). After synthesis, this mutant sequence was removed from the pUC57 vector by BamHI and AscI digestion. The liberated fragment was swapped into the place of the wild type sequence in the pUC57 *bab1* ORF vector. The full length *bab1* DBM ORF was then removed from the pUC57 vector by EcoRI and NotI digestion and subcloned into the pUMA vector. All ORF transgenes were site-specifically integrated into the *D. melanogaster* attP40 site (*Markstein et al., 2008*) on the second chromosome.

## Alignment of nucleotide and protein coding sequences

The nucleotide sequences of the wild-type *D. melanogaster* yBE0.6 CRE and scan mutant versions were aligned using the CHAOS ad DIALIGN software system (*Brudno et al., 2004*) (*Figure 2—source data 2*). The protein coding sequences for Bab homologs were aligned using the Clustal Omega multiple sequence alignment program (*Sievers et al., 2011*) (*Figure 6—source data 1*). These homologs were *D. melanogaster* Bab1 (gbAAF47439.2) and Bab2 (AAF47442.2), *D. ananassae* Bab1 (GF10081-PB) and Bab2 (XP_001956730.1), *D. willistoni* Bab1 (GK16863-PB), *D. mojavensis* Bab2 (XM_002007001.1), *Glossina morsitans* Bab1 (GMOY011079) and Bab2 (GMOY011080-RA), and *A. gambiae* Bab (AGAP006018-RA).

## Imaging and analysis of abdomens

Images of the adult *D. melanogaster* abdomen pigmentation patterns were taken with an Olympus SZX16 Zoom Stereoscope equipped with an Olympus DP72 digital camera. Specimens were prepared from 2- to 5-day-old adults. To measure the mean darkness of adult tergite pigmentation phenotypes, for each genotype four replicate images were converted to grayscale in Photoshop CS3 (Adobe). These grayscale images were opened in the Image J program (*Abràmoff et al., 2004*), and with the freehand selection tool tergite regions were specified to obtain the grayscale darkness value from that lies on a 0–255 scale. For each measurement, the % darkness was calculated as: (255-grayscale darkness)/255 × 100 (*Rebeiz et al., 2009*). From the percent darkness values, standard error of the means (SEM) were determined. Measurements and calculations provided in *Figure 5—source data 1* and *Figure 6—source data 1*.

The expression patterns of EGFP-NLS reporter transgenes were recorded as projection images by an Olympus Fluoview FV 1000 confocal microscope and software. The regulatory activities of the *yellow* 5′ sequences were imaged at ~85 hAPF, a time point when endogenous *yellow* expression occurs in the abdominal epidermis (*Camino et al., 2015*; *Jeong et al., 2006*). Representative images were selected from the replicate specimens for display in figures presenting reporter transgene expressions. All images underwent the same modifications as the control specimens using Photoshop CS3 (Adobe).

Quantitative comparisons of the levels of EGFP-NLS reporter expression were performed similar to that previously described for another CRE (*Camino et al., 2015*; *Rogers et al., 2013*; *Rogers and Williams, 2011*; *Williams et al., 2008*). In brief, for each reporter transgene in a genetic background ectopically expressing a Bab protein, EGFP expression was imaged from five independent replicate specimens at developmental stage of ~85 hr after puparium formation (hAPF) for *D. melanogaster* grown at 25°C. Imaging was done with confocal microscope settings optimized so that few pixels were saturated when EGFP-NLS expression was driven in a control background ectopically expressing the DNA-binding mutant form of Bab1 (Bab1 DBM) under the regulation of the *y-GAL4* chromosome. For each confocal image, a pixel value statistic was determined for the dorsal epidermis of the A4 and A5 segments and separately the A3 segment using the Image J program (*Abràmoff et al., 2004*). In *Figures 7* and *8*, the mean pixel values were converted into a % regulatory activity with its SEM. The activities were normalized to the activity in the background expressing the Bab1-DBM which were considered to be 100%. Measurements and calculations provided in *Figure 7—source data 1* and *Figure 8—source data 1*. Pixel values of replicate specimens are presented in *Figure 8—figure supplement 1*.

## Acknowledgements

This study benefited from stocks obtained from the Bloomington Drosophila Stock Center (NIH P40OD018537), the San Diego Drosophila Stock Center, and from Sean B Carroll. We are grateful for Michael Levine sharing the pattB-NE3 and pHB vectors to make shmiR transgenes. MJR was supported in part by the Berry Summer Thesis Institute and Honors Program at the University of Dayton. EMC and SG were supported by fellowships from the University of Dayton Graduate School. TMW and MR were supported by a grant from the National Science Foundation (IOS-1555906). MR's work on Drosophila pigmentation was supported by a grant from the National Institutes of Health (5R01GM114093-02). The funders had no role in study design, data collection and analysis, decision to publish, or preparation of the manuscript.

## Additional information

### Funding

| Funder | Grant reference number | Author |
| --- | --- | --- |
| National Science Foundation | IOS-1555906 | Mark Rebeiz<br>Thomas Michael Williams |
| National Institutes of Health | 5R01GM114093-02 | Mark Rebeiz |

The funders had no role in study design, data collection and interpretation, or the decision to submit the work for publication.

### Author contributions

Maxwell J Roeske, Conceptualization, Resources, Data curation, Formal analysis, Supervision, Funding acquisition, Investigation, Methodology, Writing—original draft, Writing—review and editing; Eric M Camino, Conceptualization, Data curation, Formal analysis, Investigation, Methodology, Writing—original draft; Sumant Grover, Data curation, Formal analysis, Investigation, Methodology, Writing—review and editing; Mark Rebeiz, Investigation, Methodology, Writing—review and editing; Thomas Michael Williams, Formal analysis, Funding acquisition, Writing—original draft, Writing—review and editing

### Author ORCIDs

Thomas Michael Williams (iD) http://orcid.org/0000-0003-1363-0637

### Decision letter and Author response

Decision letter https://doi.org/10.7554/eLife.32273.031
Author response https://doi.org/10.7554/eLife.32273.032

## Additional files

### Supplementary files

• Transparent reporting form
DOI: https://doi.org/10.7554/eLife.32273.029

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
