## [Decision Letter]

Thank you for submitting your article "Cis-regulatory evolution integrated the Bric-à-brac transcription factors into a novel fruit fly gene regulatory network" for consideration by *eLife*. Your article has been reviewed by three peer reviewers, and the evaluation has been overseen by Patricia Wittkopp as the Senior and Reviewing Editor. The reviewers have opted to remain anonymous.

The reviewers have discussed the reviews with one another and the Reviewing Editor has drafted this decision to help you prepare a revised submission.

Summary:

In this manuscript, the authors study how an ancient transcription factor became integrated into a gene regulatory network (GRN) underlying the novel sexually dimorphic abdominal pigmentation in *Drosophila* species. Specifically, the authors elucidate the regulatory linkage between the two key TFs in this GRN, the Bab paralog pair, and the effector pigmentation pathway gene, yellow. They address two main questions: 1) How and when was Bab incorporated into the dimorphic pigmentation GRN? 2) To what extent did gene duplication, protein coding sequence and CRE evolution contribute to the derived trait? The experiments conducted to address these questions were well-designed and the paper is well written. We have a few major requests for revision.

Essential revisions:

1) Please provide a more complete introduction to the GRN under study. For example, Abd-B is mentioned throughout the text, but the exact role of this TF in this network is rather unclear. Either elaborating Figure 8, or having a diagram of the whole GRN in (or as) Figure 1 will better contextualize the Bab-yellow regulatory linkage and will help orient the readers, when discussing the different components of the GRN in the text.

2) Tone down the language related to the evolutionary conservation of Bab functions after duplication. Over-expressing a gene with the GAL4-UAS system produces protein levels that are not in the physiological range. So, it is likely that the amount of bab in epidermal cells is saturating. Indeed, this might be the reason why the level of suppression of yBE0.6 is so similar for all bab homologs (Figure 6). With the assay used, it cannot be claimed that the function of these genes has been conserved and that the suppressive capabilities of homologs are similar. For example, DNA binding affinity might have changed, but this wasn't visible because of the over-expression of these proteins, which might pass a threshold. To be clear, we are not asking for additional experiments to address this point given the time and effort required. Rather, we are asking you to solve this issue by softening the conclusions. A sequence comparison showing there have not been radical amino acid changes in Bab homologs would also be helpful to include. The following three sentences, for example, should be modified: "by these ectopic expression assays, we find that not only are the Bab paralogs sufficient to suppress pigmentation when ectopically expressed, but that their suppressive capabilities are similar"; "These results suggest that the ability of the Bab1 and Bab2 proteins to regulate dimorphic pigmentation did not require noteworthy changes in their protein coding sequences"; "These outcomes are consistent with the interpretation that the general functional capability of these proteins was ancient and has been generally conserved during the 100 million years or more since the gene duplication event occurred".

3) Refine your conclusions regarding Bab evolution. The regulation of this enhancer by Bab (bab1+bab2) is probably a derived condition, and the regulation of yellow by Bab evolved at roughly the same time as the regulation of bab by Abd-B (another regulatory linkage that contributes to sexually dimorphic pigmentation). However, this conclusion comes with the caveat that taxon sampling in this study is very limited. It remains possible that the bab-yellow linkage is more ancient but has been lost/reduced in some clades.

---

## [Author Response]

Essential revisions:1) Please provide a more complete introduction to the GRN under study. For example, Abd-B is mentioned throughout the text, but the exact role of this TF in this network is rather unclear. Either elaborating Figure 8, or having a diagram of the whole GRN in (or as) Figure 1 will better contextualize the Bab-yellow regulatory linkage and will help orient the readers, when discussing the different components of the GRN in the text.

We made changes to help clarify the role of Abd-B. As suggested, we created a new Figure 1 that represents what we consider to be the contemporary understanding of the *D. melanogaster* abdominal tergite pigmentation GRN. A key part of this GRN is the role of *Abd-B* as an upstream activator of melanic pigmentation genes such as *yellow* and *tan*. We also modified a sentence in the Introduction to now read: “Within the GRN controlling this trait, the Hox gene *Abd-B* plays an important role in activating the expression of terminal enzyme genes that are required for pigment formation (Figure 1).”

2) Tone down the language related to the evolutionary conservation of Bab functions after duplication. Over-expressing a gene with the GAL4-UAS system produces protein levels that are not in the physiological range. So, it is likely that the amount of bab in epidermal cells is saturating. Indeed, this might be the reason why the level of suppression of yBE0.6 is so similar for all bab homologs (Figure 6). With the assay used, it cannot be claimed that the function of these genes has been conserved and that the suppressive capabilities of homologs are similar. For example, DNA binding affinity might have changed, but this wasn't visible because of the over-expression of these proteins, which might pass a threshold. To be clear, we are not asking for additional experiments to address this point given the time and effort required. Rather, we are asking you to solve this issue by softening the conclusions. A sequence comparison showing there have not been radical amino acid changes in Bab homologs would also be helpful to include. The following three sentences, for example, should be modified: "by these ectopic expression assays, we find that not only are the Bab paralogs sufficient to suppress pigmentation when ectopically expressed, but that their suppressive capabilities are similar"; "These results suggest that the ability of the Bab1 and Bab2 proteins to regulate dimorphic pigmentation did not require noteworthy changes in their protein coding sequences"; "These outcomes are consistent with the interpretation that the general functional capability of these proteins was ancient and has been generally conserved during the 100 million years or more since the gene duplication event occurred".

We added an amino acid alignment for diverse Bab homologs as a Figure 2—source data 1.

We changed the sentence "Thus, by these ectopic expression assays, we find that not only are the Bab paralogs sufficient to suppress pigmentation when ectopically expressed, but that their suppressive capabilities are similar.” to “These data show that Bab1 and Bab2 are both potent suppressors of pigmentation in two very different regimes of ectopic expression.”

We changed the sentence “These results suggest that the ability of the Bab1 and Bab2 proteins to regulate dimorphic pigmentation did not require noteworthy changes in their protein coding sequences.” to “These results suggest that the ability of the Bab1 and Bab2 proteins to regulate dimorphic pigmentation did not require the evolution of an altogether new biochemical capability.”

We changed the sentence “These outcomes are consistent with the interpretation that the general functional capability of these proteins was ancient and has been generally conserved during the 100 million years or more since the gene duplication event occurred.” to “These outcomes are consistent with the interpretation that the general functional capability of these proteins was ancient and has been conserved during the 100 million years or more since the gene duplication event occurred. However, these experiments cannot rule out the possibility that changes have occurred in the *bab* protein coding sequences that cause incremental differences in the ability to regulate dimorphic pigmentation. In fact, the further reduced pigmentation caused by the Bab1 orthologs compared to those for Bab2, and *A. gambiae* Bab suggests that such incremental enhancements have indeed occurred (Figure 6).”

We made an additional change in the Abstract to further tone down the language. Specifically, we changed “Here we show that Bab gained a role in sculpting sex-specific pigmentation through the evolution of binding sites in a CRE of the pigment-promoting *yellow* gene and without any noteworthy changes to Bab protein coding sequences.” to “Here we show that the ancestral transcription factor activity of Bab gained a role in sculpting sex-specific pigmentation through the evolution of binding sites in a CRE of the pigment-promoting *yellow* gene.”

3) Refine your conclusions regarding Bab evolution. The regulation of this enhancer by Bab (bab1+bab2) is probably a derived condition, and the regulation of yellow by Bab evolved at roughly the same time as the regulation of bab by Abd-B (another regulatory linkage that contributes to sexually dimorphic pigmentation). However, this conclusion comes with the caveat that taxon sampling in this study is very limited. It remains possible that the bab-yellow linkage is more ancient but has been lost/reduced in some clades.

We changed the sentence “Collectively, these results suggest that the direct Bab regulation of *yellow* evolved specifically within the dimorphic body element, coincident with the evolution of the dimorphic pigmentation trait.” to read as “These results can be explained by two evolutionary scenarios. […] In the future, deeper taxon sampling in *Sophophora* may shed light on which of the two scenarios reflects the true evolutionary history for this dimorphic pigmentation trait.”

We think this clarification fits well with our soon to follow Discussion section on “Bab and its history in a pigmentation gene regulatory network.”